

# "More poison than words can describe": What did people die of after the 1783 Laki eruption?

Claudia E. Wieners[1*] and Guðmundur Hálfdanarson[2]

[1]Institute for Marine and Atmospheric research, Utrecht, Utrecht University, Netherlands
[2]Faculty of Philosophy, History and Archaeology, University of Iceland
[*]Corresponding author, c.e.wieners@uu.nl

**Abstract.** The 1783 Laki eruption in Iceland was followed by an almost 20% population loss, traditionally attributed to famine (caused by fluorine poisoning of livestock) and contagious diseases. From the 1970s, hypotheses were formed that fluorine poisoning might have contributed to human mortality in Iceland, and air pollution might have caused excess deaths both in Iceland and Europe. Using historical documents including parish registries, we find that regional and temporal patterns in Icelandic excess mortality can be satisfactorily explained by hunger and disease, when other factors such as the availability of secondary food sources (fishing, food aid) are taken into account. In contrast, the timing and estimated concentrations of air pollution do not match observed excess mortality, and observed symptoms and estimated human fluorine uptake do not suggest large-scale fluorosis in humans. We therefore conclude that the evidence for significant direct contributions from pollution to human mortality is weak.

## 1 Introduction

The flood basalt eruption of the Lakagígar fissure in 1783, commonly referred to as Laki eruption, was arguably the worst natural disaster in the history of Iceland. Although the eruption did not take human lives directly through lava, tephra fall or similar phenomena, in the words of a local pastor "more poison fell from the sky than words can describe" (Steingrímsson, 1788), and acid rain and fluorine-rich ash caused a massive loss of livestock (Pétursson et al., 1984; Rafnsson, 1984) leading to famine. The Icelandic population dropped from 49609 at the beginning of 1783 to 40381 at the beginning of 1786 (Jónsson et al, 1997) due to excess mortality and a reduction in childbirths. Until the 1980s, most historians attributed human excess mortality entirely to hunger and disease (Pétursson et al., 1984; Hálfdanarson, 1984).

In the 1990s, Grattan and co-workers analysed parish data in England and France and found significant excess mortality in the late summer 1783 and the following winter, which they attributed to air pollution blown over from Iceland (Grattan et al., 2003, 2005). Their studies sparked interest in the possible health impact of future Laki-type eruptions (Schmidt et al., 2011; Heaviside et al., 2021) and led to the suggestion that pollution also helped increase excess mortality in Iceland itself:





*"Icelandic records clearly suggest that air pollution contributed directly to increased mortality rates, with northern regions repeatedly exposed to a low-level sulfuric acid haze suffering much higher death rates than other regions"* (Schmidt et al.,
2011). However, no attempt has been made to carefully test this hypothesis combining written sources, mortality statistics, and environmental (modelling) data.

Another suggested cause of human mortality in Iceland (but not beyond) is fluorine poisoning, which, as mentioned, was prevalent in livestock. Blong (1984) suggests that the symptoms in humans described by Stephensen (1785), who visited the region in 1784, might be "partly" attributed to fluorosis. Weinstein (2005), discussing the question whether the saga figure
Egill Skallagrímsson suffered from chronic fluorosis, refers to the careful observations of the local pastor Jón Steingrímsson Steingrímsson (1788) as potential evidence for human fluorine poisoning during the eruption. D'Alessandro (2006) cites the Laki eruption as the only known example of potential human fluorine poisoning due to an active volcanic eruption and, referring to Friðriksson (1983), writes that *"Jón Steingrímsson [...] tells us also that some of the people [...] developed the same bone and teeth deformations like the animals."* However, few attempts have been made to conduct a systematic differential diagnosis
with scurvy (vitamin C deficiency) or estimate the probable level of human fluorine intake.

We suspect that the hypotheses of human mortality due to air pollution and fluorine poisoning caused by the Laki eruption have been insufficiently scrutinised and potentially over-emphasised, including in popular science literature (Witze, 2016). In this article, we conduct a critical review of written records, Icelandic mortality data, studies of contemporary pollution events and modelling studies, to investigate whether air pollution and fluorine poisoning made a direct, significant contribution to
excess mortality in Iceland and beyond.

One aspect we do not study in depth is the eruption's climate impact. Stratospheric sulphate aerosol ejected by the volcano reflected incoming sunlight and led to Northern hemisphere cooling (Stevenson et al., 2003; Highwood and Stevenson, 2003; Chenet et al., 2005; Oman et al., 2006b; Zambri et al., 2019a, b). It might have upset regional weather patterns and caused or exacerbated famine in the Nile region due to drought (Oman et al., 2006a). If caused by the eruption, these anomalies would contribute significantly to its death toll. However, regional climate anomalies can only be attributed in a probabilistic
sense: Even if Laki-style eruptions increase the likelihood of certain weather events, it is impossible to prove that particular outcomes in 1783-84 were indeed caused by the eruption. For example, the cold winter 1783-84 in central Europe may have been exacerbated by larger-scale cooling, but may also have been (partly) related to El Niño (D'Arrigo et al., 2011), although El Niño itself may have been influenced by the eruption (Pausata et al., 2011). The hot summer in 1783 in central Europe likely
occurred despite, and not because of, the eruption (Zambri et al., 2019b).

Our study is structured as follows. After sketching the socio-economic situation of Iceland and the environmental impacts of the eruption (sect. 2), we describe the data and methods used (sect. 3). In sect. 4, we give an overview over Icelandic mortality data, including spatial and temporal patterns, in the crucial period. In sect. 5, we investigate whether famine and a famine-induced rise in endemic disease satisfactorily explains the available Icelandic data. After a brief review on mortality in Europe
(sect. 6), we investigate in sect. 7 the possibility of human death by air pollution, using written records and modelling studies. Similarly, in sect. 8, we use written sources and our own estimates of fluorine intake to assess the possibility of wide-spread lethal fluorosis in humans.



## 2 Socio-economic and environmental situation

### 2.1 Socio-economic background

At the time of the Laki eruption, Iceland had been a province in the Danish-Norwegian Monarchy for over four hundred years. Local Icelandic officials received instructions from Danish institutions in Copenhagen. In Iceland there were two interlocking administrative systems, one civil and the other religious. The basic units of the former were communes (*hreppur*), which handled poor relief, and counties (*sýsla*, pl. *sýslur*); fig. 1a). The latter system consisted of parishes (*sókn*) and deaneries (*prófastsdæmi*). Both systems were used for raising data for the royal authorities in Copenhagen. The boundaries of counties

and deaneries were very similar, but not identical, which has to be corrected for when comparing different types of statistics. In particular, the counties Gullbringusýsla and Kjósarsýsla were one deanery and appear as one unit in demographic data, and Hnappadalssýsla was split between Snæfellsnes- and Mýrasýsla.

Economically, Iceland prior to 1783 was largely a subsistence farming community with fishing as an important secondary activity (Gunnarsson, 1983). The population was around 50000 people who lived along the coast, in valleys and low-lying

plains (fig. 1a). Animal husbandry (cows and sheep) was the dominant occupation, and hay the main crop. Dairy products accounted for half of the Icelanders' calory intake (Karlsson, 2000). Fishing was done in open rowing boats which could not venture out far to sea, hence fishermen could not follow the fish. The best fishing grounds, with fishing opportunities in late winter and spring, were in the west, particularly Gullbringu- and Snæfellsnessýsla (fig. 1b). Farmers and farmhands from other regions often travelled to the fishing regions to participate in winter fishing or barter dried fish for farming products.

The number of cow equivalents (6 ewes being traditionally valued like 1 cow) can serve as a rough index of farming-based food production. Not surprisingly, districts with good access to fishing - particularly the western peninsulas - tended to have less livestock (fig. 1b). The figure also suggests, however, that the eastern regions, especially the Múlasýslur, had less cow equivalents than regions with similar fishing access further west. This may be the expression of fluctuations, for example due to relatively harsh weather in the preceding years (Guðjónsson, 2010).

Contact with the outside world was mostly limited to Danish merchants, who served the roughly 20 trade posts (fig. 1a). Main export goods were dried fish, mutton, and wool products; imports included grain, but also wood, iron and fishing lines. The trade posts were the only locations where Danish food aid could be delivered. Such aid did not arrive before July 1784, and even then in meagre quantities (Wieners, 2020).

### 2.2 Environmental impacts of eruption

The eruption lasted from June 8th, 1783, to February 7th, 1784, though the activity was largest in the first months (Thordarson and Self, 1993). Thanks to its location in the uninhabited inland of southern Iceland (fig. 1a), the eruption did not directly kill people, but when the lava flowed into inhabited areas, 42 farmsteads and cottages ($\approx 0.8\%$ of all Icelandic farms) were given up because of lava flows, inundation by dammed-up rivers, tephra fall and sand storms; another 19 were abandoned for other reasons including loss of livestock (Guðbergsson and Theodórsson, 1984). About 500 out of 1964 inhabitants fled

Vestur-Skaftafellsýsla county, mostly to the west (Gunnlaugsson, 1984a).



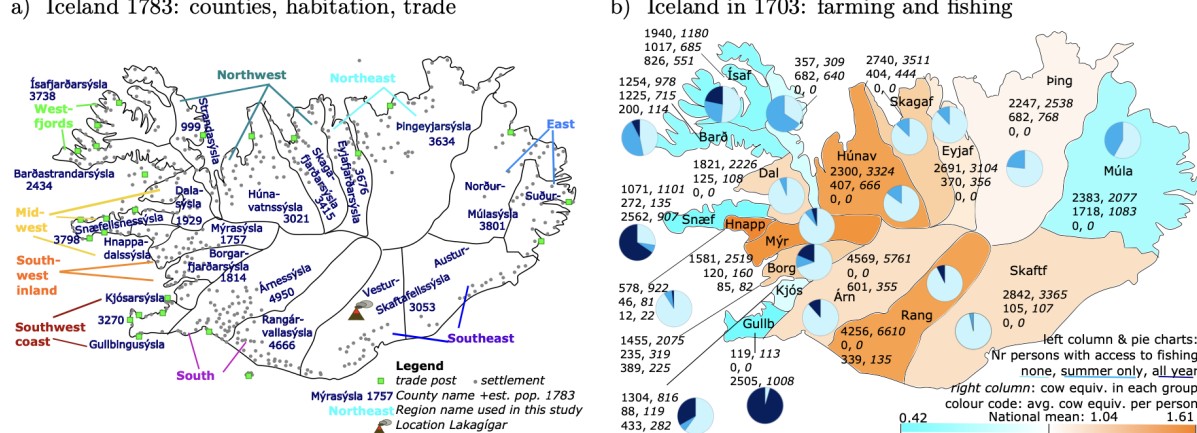

**Figure 1.** Socio-economic background in Iceland. a) Counties (*sýslur*), distribution of the population (qualitative indication only), location of Danish trade posts (Gunnarsson, 1983), position of the Laki fissure, and regions (groups of counties) used in the analysis. Numbers are the 1783 population estimated using the 1769 census data and birth and death data. b) Colour code: cow equivalents per person; 6 ewes count as one cow. The colour code is such that the darkest orange (blue) correspond to the highest (lowest) availability of cow equivalents, while white depicts the national mean. Pie charts denote the fraction of the population with no access to fishing, summer-only access, or year-round (i.e. also winter) access. Numbers are: left column, the number of persons with access to no (top), summer-only (middle), or all-year (bottom) fishing, while the right colunm shows the number of cow equivalent owned by each group. Data taken from Magnússon (1786). The data is also presented in table form in SM2.

As opposed to local lava flows, volcanic haze and ash fall affected most of Iceland (Thordarson, 1995). The Lakagígar magma was rich in SO2, which within a few days reacts in the atmosphere to form droplets of sulphuric acid (H2SO4) (Stevenson et al., 2003), leading to the formation of a thick, "dry", acidic haze. Roughly 20% of the $\approx 120 Mt$ (Megatonnes) of SO2 slowly degassed from the lava streams and mostly affected southern Iceland, while 80% was carried by eruption columns into the high

troposphere and lower stratosphere (9-13km height) (Thordarson and Self, 2003), got transported by high-level winds, and was partly re-introduced to the lower troposphere by subsidence, both in Iceland and in mainland Europe (Thordarson, 1995). Low-altitude haze was observed, on and off, in all parts of Iceland during the summer and autumn of 1783, and in mainland Europe from the second half of June to September-December (depending on location) (Thordarson, 1995; Grattan and Brashay, 1995; Thordarson and Self, 2003; Halldórsson, 2013). The haze caused massive damage to the vegetation in Iceland, including grass

(Steingrímsson, 1788; Thordarson, 1995), which was all-important for animal husbandry, but also secondary food sources like lyme grass, berries, and Icelandic moss, were much diminished due to the eruption (Steingrímsson, 1788; Pétursson et al., 1984). In Europe, some plant damage was observed but did not lead to harvest failure (Halldórsson, 2013).

Fine ash was produced by lava fragmentation during explosive episodes and spread over an area of $200000 km^2$, including most of Iceland, except the extreme west and northwest (Thordarson and Self, 2003). The ash carried fluorine. It has been

estimated that $\approx 8 Mt$ of fluorine was released at the vents (Thordarson and Self, 1996), of which $9 \times 10^7 kg$ got adsorbed by





fine ash and deposited along with it (Thordarson and Self, 2003). Spread over an area of $200000km^2$, this yields $450mg/m^2$ on average [1].

Symptoms associated with fluorine poisoning of livestock were reported from nearly all over Iceland, except in the northwest (Pétursson et al., 1984), consistent with the spread of fine ash. Animals, especially sheep, started to die within two weeks after

the onset of the eruption in nearby parishes (Steingrímsson, 1788); elsewhere it took several months for livestock to die (Pétursson et al., 1984). Even before animals died, a drop in milk production to one-half or less was noticed immediately after the arrival of the haze, in Vestur-Skaftárfellssýsla (Steingrímsson, 1788), but also in the North. Lack of fodder exacerbated the loss of livestock (Wieners, 2020, SM).

Wild animals, such as fresh water fish and birds, were also reduced (Steingrímsson, 1788; Pétursson et al., 1984), while there

is no indication that marine fish were affected.

## 3 Data and Methods

Through the 18th century, the Danish administration showed growing interest in its distant Icelandic dependency, reflecting new centralizing tendencies in the capital. This interest appeared in various efforts to collect information on Iceland's economy, demography, and natural conditions, because if the state was to control and develop the periphery, the authorities had to

understand it. For this reason, we have access to a wealth of information on various aspects of Iceland's demography and economy during the years of the Laki eruption, although there are, as in most historical statistics, gaps in the available data and uncertainties on its accuracy.

In addition to demographic data explained below, we made use of letters by local officials Gunnlaugsson and Rafnsson (1984), which were summarised in English in the supplementary material of Wieners (2020) (henceforth Wieners20SM). A

few additional reports were used; these are listed in the reference list.

### 3.1 Icelandic demographic data

When it comes to mortality data from the late 18th century, we have primarily two sets of sources. First, there are general vital statistics from the period – that is, raw numbers on deaths and births, broken down on county level – both in print and in unpublished reports preserved in the National Archives of Iceland (NAI) in Reykjavík. For this data, we used a set of

tables published in the 18th-century Icelandic journal, *Rit þess íslenzka lærdómslistafélags* (The Proceedings of the Icelandic Learned Society), as basis (RÞÍL, 1781-1798, VI, 249-263; VII, 251-269; VIII 271-274; IX, 287-289). They cover the period from around 1770 to the end of the 1780s, and were compiled by two Icelandic officials, using reports composed by parish priests every year (Stephensen and Sigurðsson, 1854, II, 226-227). Obvious mistakes and gaps in the tables were corrected by comparing them to the existing original reports from the two Icelandic bishops, preserved in the National Archive of Iceland.

Two remaining instances of incomplete data (Rangárvallasýsla 1776, Dalasýsla 1774) were filled by assuming that birth and

---

[1]Thordarson and Self (2003) give $500mg/km^2$, which contains a typo in the unit.





death rates developed in parallel with neighbouring counties:

$$D(C,Y) = \frac{(D(C,Y-1)+D(C,Y+1))(D(C-1,Y)+D(C+1,Y))}{D(C-1,Y-1)+D(C+1,Y-1)+D(C-1,Y+1)+D(C+1,Y+1)} \qquad (1)$$

where $D$ is the number of deaths (or births), $C$ and $Y$ the county and year for which data is missing, and $C\pm1$ the neighbouring counties.

Second, every pastor in Iceland was required to keep a register of the tasks he performed, including burials and baptisms in their parish (Prestsþjónustubækur, 1782-87). The information provided varies, and it changes over time; invariably, the pastor registered the names of the deceased persons (indicating their gender) and the time of burial, but most of the registers also provide additional data, including the actual dates and presumed causes of death, the ages of the deceased, where they lived, and their social status at death. We used all available parish registers from the period 1782, or right before the eruption, to 1787,

when the crisis following the eruption was over. Using them, we were able to create a database of almost five thousand registered burials, providing a detailed and individualised picture of the demographic crisis in 1784–1786. Additional explanations are given in the Supplementary Material (SM1). Unfortunately, no registry data are available for Skaftafellssýsla, the county closest to the volcano. Also, information from some parishes is missing for 1782 and/or 1783. In the northern counties, recording causes of death only become mandatory from 1785, leading to a high number of "unspecified" causes of death.

Population counts were only performed sporadically. The closest complete census prior to the eruption took place in 1769; the data is available at parish level (Jónsson et al, 1997). Minor mistakes were corrected using (Manntalstöflur, 1769). To be able to compare census data with the mortality data, we grouped parishes according to the (county) boundaries used in the county-level data.

All three data sets are available online as Supplementary data.

## 3.2    Demographic data: Estimation methods and index definition

To analyse demographic data, we used the following definitions:

*Estimated population (county level):* As the last complete census prior to 1783 was performed in 1769, we estimated the county-level population up to 1783 by adding births and subtracting deaths. Population is given at the beginning of each calender year. The method ignores migration across county borders, therefore it is unreliable after 1783. (Migration out of

Iceland as a whole was negligible.)

*Death rates (county level):* Death rates for 1770-83 were computed by dividing the number of deaths by the estimated population at the beginning of the year. For 1784-85, the estimated population at the beginning of 1783 was used because migration prevents us from forming reliable population estimates.

*Relative mortality (county level):* Relative mortality is computed by dividing the number of deaths in one county for a year

of interest (1783, 1784, 1785) by the average number of deaths 1773-82.

*Normalising parish-level data:* Parish-level data were aggregated on county level. Since the coverage is incomplete, the number of deaths in any given category $G$ (e.g. people who died of hunger) were extrapolated to county level as follows:

$$D(C,G,Y) = D(P_C,G,Y) * D(C,Y)/D(P_C,Y) \qquad (2)$$





where $D(C, G, Y)$ are the estimated deaths in category $G$ in county $C$ and year $Y$; $D(P_C, G, Y)$ the number of deaths in category $G$ and year $Y$ counted in parish-level data for all available parishes within county $C$, $D(C, Y)$ the total number of deaths in county $C$ and year $Y$ (taken from county-level data) and $D(P_C, Y)$ the number of all deaths counted in the available parish-level data of that county.

*Migrants' county of origin:* Pastors often denoted the region of origin of people they buried, but not always at county level. When pastors listed a place (farm, parish, *hreppur*) of origin, we tried to locate it and assign the county. In case two places of that name occur in different counties, we assumed that the one nearer by was meant, if no such distinction could be made or the location was unidentifyable, the region of origin was deemed "unspecified". Migrants "from the eruption" were assumed to come from the county containing the volcano, Vestur-Skaftafellssýsla. Migrants "from the east" who died in the south and west were likewise assumed to come from to Vestur-Skaftafellssýsla, which provided far more migrants than all other counties in the east and south of Iceland. Migrants "from the north" were counted by adding 1/4 to each of the four counties of the northern quarter (Húnavatns- to Þingeyjarsýsla). The four counties had similar populations before 1783. Migrants from unspecified locations were then distributed as follows: For any county receiving immigrants $C_r$ and any source county $C_s$ and year $Y$, we assume

$$M(C_r, C_s, Y) = M_0(C_r, C_s, Y) + M_u(C_r, Y) * \frac{M_0(C_r, C_s, Y)}{\sum_{C'_s} M_0(C_r, C'_s, Y)} \tag{3}$$

where $M(C_r, C_s, Y)$ is the final number of immigrants from $C_s$ to $C_r$ in year $Y$, $M_0(C_r, C_s, Y)$ the number of such migrants already specified, and $M_u(C_r, Y)$ the number of migrants of unspecified origin buried in $C_r$. The sum $\sum_{C'_s} f(C'_s)$ is the sum over all counties $C'_s$ of some function $f$. Unless specified otherwise, county-level burial data was used without correcting for migration. The cases $C_s = C_r$ and $C'_s = C_r$ are included (within-county migration).

*Smoothed baseline for animal ownership:* As livestock ownership in 1703 (fig. 1b) may have been affected by prior weather events, we define a "corrected" baseline for livestock ownership by assuming that people with no access to fisheries, summer-only and winter access owned 1.22, 0.85 and 0.46 cow equivalents per person, respectively (i.e., the national average over each fishing category). In other words:

$$E_{corr}(C) = \frac{\sum_{C'} E_{no}(C')}{\sum_{C'} P_{no}(C')} P_{no}(C) + \frac{\sum_{C'} E_{su}(C')}{\sum_{C'} P_{su}(C')} P_{su}(C) + \frac{\sum_{C'} E_{wi}(C')}{\sum_{C'} P_{wi}(C')} P_{wi}(C) \tag{4}$$

where $no$, $su$, and $wi$ stand for no access to fisheries, summer-only access and winter access, $P_x(C)$ the population in county $C$ and fishing category $x$, and $E_x(C)$ for the total number of cow equivalents owned by the inhabitants of county $C$ and fishing category $x$. The sum $\sum_{C'} f(C')$ is the sum over all counties $C'$ of some function $f$.

## 4 Mortality in Iceland 1783-1786





| year | 1782 | 1783 | 1784 | 1785 | 1786 | 1787 | mean |
|---|---|---|---|---|---|---|---|
| population | 49611 | 49609 | 49753 | 45428 | 40381 | 39190 | |
| deaths (%) | 1231 (2.5) | 1227 (2.5) | 5429 (10.9) | 5649 (12.4) | 2128 (5.3) | 920 (2.4) | 1522 |
| births (%) | 1229 (2.5) | 1371 (2.8) | 1104 (2.2) | 602 (1.3) | 937 (2.3) | 1220 (3.1) | 1501 |

**Table 1.** Population, deaths and births in Iceland, 1782-87. Population is given for the beginning of the corresponding year. The percentages of deaths are number of deaths in a given year divided by the population at the beginning of that year; similar for births. The mean values are for 1750-1800. Data from Jónsson et al (1997).

### 4.1 Mortality at county level

As can be seen from table 1, the Laki eruption was followed by a large mortality peak in 1784 and 1785, as well as a dip in births. However, no excess mortality can be discerned for the eruption year, 1783. In the two peak years, about 11,000 persons

died in Iceland, which amounted to excess mortality of around 8,000 persons, given that the average annual mortality in 18th-century Iceland was just over 1,500 (Jónsson et al, 1997). The excess mortality during these two years amounted, therefore, to one-sixth of the pre-crisis population.

Mortality remained elevated in 1786, but this was mainly caused by a smallpox epidemic, mostly unrelated to the Laki eruption. It started in the late autumn 1785 and spread around the country in the following year. Population decline was

increased by a decrease in births, particularly in 1785 (Vasey, 1991), a common phenomenon during famines.

Mortality was unevenly distributed across the Icelandic counties. In fig. 2a,b, two measures for the severity of the crisis are presented. The first measure is the relative mortality with respect to 1773-82. The second measure is an estimated death rate, computed by dividing the number of deaths in the years of interest (1783, 1784, 1785) by the estimated population at the beginning of 1783, i.e., at pre-famine levels. This reference is chosen because migration makes it difficult to estimate population

at the end of 1784 (sect. 3.2). Both measures show that 1783 had below-average mortality in most of the country, except in the three easternmost deaneries, particularly in Þingeyjarsýsla, where it was about 50% above average. In the north-eastern part of Iceland plus Ísafjarðarsýsla in the Westfjords, mortality was highest in 1784, while in the south-west it was highest in 1785. In the following, we focus on the years 1784–1785.

Both mortality measures show that the Westfjords (Ísafjarðar- and Barðarstrandarsýsla) and the south (Árnes- and Rangár-

vallasýsla) escaped comparatively lightly, though not unscathed (the two-year mortality was about 2.5 times the normal value), whereas all the counties in North Iceland (from Húnavatns- to Þingeyjarsýsla) had very high mortality in 1784–1785, Þingeyjarsýsla being worst hit with a mortality about 8.5 times normal. The southwest and west – from Gullbringu- and Kjósarsýsla to Snæfellsnessýsla – had relatively high excess mortality too, but here the two measures disagree considerably about the relative severity of the crisis. When using deathrates, Gullbringu- and Kjósarsýsla were the hardest hit of all the counties in Iceland,

and Snæfellsnes- and Mýrasýsla were almost as badly affected as Þingeyjarsýsla. When using relative mortality, the worst-hit counties were Þingeyjar- and Húnavatnssýsla in the north, followed by Mýrasýsla in the west, while relative mortality in Gullbringu- and Kjósarsýsla was close to the national average. In Skaftafellssýsla, the county where the Laki fissure is located,



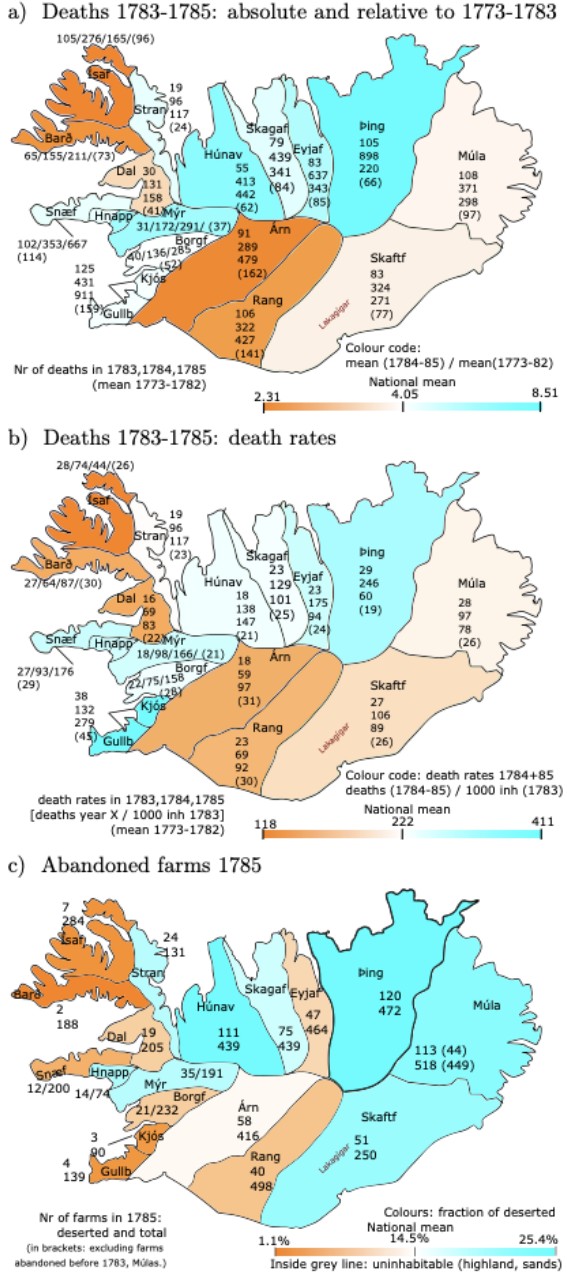

**Figure 2.** Regional distribution of death and abandoned farms. a) Colours depict relative mortality 1784-85, i.e. the mean over 1784-85 divided by the mean over 1773-82. Numbers are, from top to bottom: Deaths in 1783, 1784, 1785, and the mean 1773-1782 (in brackets). b) Colours depict the two-year death rates 1784-85, i.e. deaths 1784-85 divided by the estimated population at the beginning of 1783. Numbers, from top to bottom, are the deathrates for 1783, 1784, 1785 (all with respect to the population at the beginning of 1783!), and, in brackets the mean death rate 1773-82 (mean over death in year Y / population in the beginning of year Y). c) Colours depict the fraction of abandoned farms 1785. Numbers show the number of abandoned (top) farms and the total number of farms (bottom). For Múlasýsla, numbers in brackets are results after excluding farms that had been abaondoned before 1783. In all plots, the colour bar is tuned such that the darkest orange (blue) colour represents the least (worst) affected county, and white the weighted national average. The data is also presented in table form in SM2.





mortality was slightly below the national average according to both measures. However, in the parishes nearby the volcano, i.e. the western part of Skaftafellssýsla, population fell significantly through out-migration to the west (Gunnlaugsson, 1984a).
As mortality data is based on burials, persons dying outside their parish were registered in the location where they died, not in their home parish.

The discrepancies between the mortality measures obviously comes from the fact that the denominator (estimated 1783 population in deathrates vs 1773-82 deaths in relative mortality) were differently distributed among counties. This can be seen from the fact that 1773-82 average death rates differed strongly (fig. 2b), with values ranging from 19 deaths per 1000
inhabitants per year (Þingeyjarsýsla) to 45 deaths per 1000 inhabitants per year (Gullbringu- and Kjósarsýsla). Snæfellsnessýsla also had rather high values, 32 deaths per 1000 inhabitants per year. Birthrates for 1773-82 show roughly similar discrepancies (not shown).

Unless systematic reporting errors are to blame, one possible explanation for the discrepancy is that people did not always die in the county where they were registered to live. This affected, in particular, the fishing regions in the southwestern and western
parts of the country (Gullbringu- and Snæfellsnessýsla and the coastal parishes in Árnessýsla), where seasonal fishermen from the inland parishes gathered in the late winter. These regions may also have attracted people on the margins of society, who were officially registered as inhabitants of their parish of origin in the 1769 census but would nonetheless contribute to the births and deaths of the parish where they actually lived, leading to higher apparent birth and death rates – especially the latter. Indeed, Gullbringu- and Kjósarsýsla was the only region with higher death rates than birth rates in 1773-82. Given these
discrepancies, relative mortality might be a more reliable measure of the (relative) severity of the crisis, because probably at least some of the systematic inconsistencies between population and death records affect counts for 1773-82 and 1783-85 in a similar manner.

Farm abandonment (fig. 2c) can provide an additional rough indicator of depopulation. The data (Rafnsson, 1984) comes with caveats: First, larger farms often contained two or more separate households, and the data do not indicate cases where only
part of the farm was given up. Second, it appears that in most districts, both the main farms (*lögbýli*) and crofts (*hjáleigur*) were counted, while in the primary fishing districts (Gullbringu-, Kjósar- and Snæfellsnessýsla), only the main farms were counted. Third, not all farms counted as abandoned were necessarily given up due to the eruption and its aftermaths. In particular, in Suður-Múlasýsla (the southern part of Múlasýsla), the county commissioner (*sýslumaður*) wrote in 1786 that of 98 abandoned farms there, 29 had been given up in 1781-86, while the others had been abandoned during the famine of 1754-60 or much
earlier. In contrast, the number of deserted farms in Skaftafellssýsla (51) hardly exceeds the 42 farms given up after being physically damaged by the eruption (sect. 2.2).

Fig. 2c suggests, once more, that the Westfjords escaped comparatively lightly, and most of the northern districts were hit hard. Skaftafellssýsla lost a substantial number of farms to the eruption. In the west, Mýra- and Hnappadalssýsla, both regions with less access to fishing, had the most deserted farms, while Snæfellsnes-, Gullbringu- and Kjósarsýsla only had few.
However, counting only main farms may lead to lower estimates in farm abandonment, because farmers on main farms were typically wealthier and less likely to give up their farm. But it may also be that even the number of deserted crofts was low





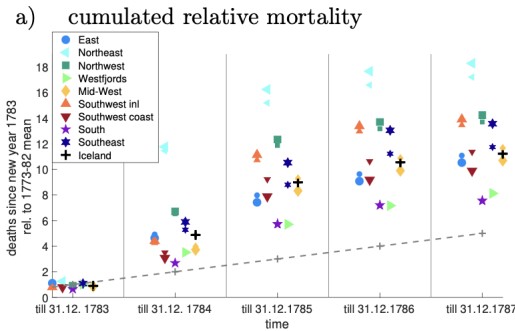 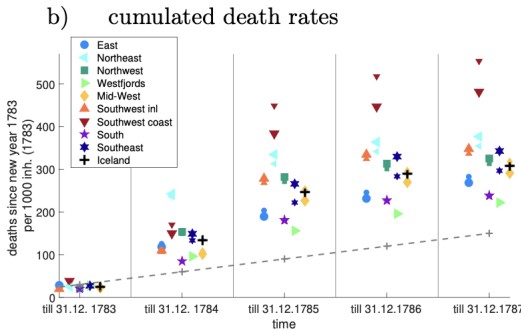

**Figure 3.** Migration and mortality. a) Cumulated number of deaths, divided by the normal (1773-82 mean) value, for each region. For example, a value of 5 in December 1785 means that from January 1st 1783 till December 31st 1783, 5 times as many people died as would normally die in one year. Regions are defined in fig. 1a. b) as a, but for death rates rather than relative deaths. For example a value of 200 in December 1785 means that 200 out of 1000 persons alive on January 1st, 1783 died by December 31st, 1785. The straight dashed line depicts normal (1773-82 whole-country) death rates. In both plots, small (big) symbols denote values without (with) correction for migration.

in these regions due to immigration. Data from around 1795 and 1802-06 suggests a shift from (hard-hit) farming regions to fishing regions (Rafnsson, 1984).

Using parish registers, one can roughly correct death counts for migration as the pastors often mentioned if a person came
from outside of the parish. In regions where pastors only gave sketchy information, especially in the north, immigrants are undercounted. No usable registers from Skaftafellssýsla exist for this period, hence immigrants are not counted there at all. However, the north and Skaftafellssýsla were mostly emigration regions, as can be seen in frequent references in the registers of the fishing districts in the west and southwest to refugees and vagabonds from these regions. Fig. 3 suggests that inhabitants of the north and in particular Skaftafellssýsla experienced higher mortality than raw data suggests, because people left these
districts and died elsewhere. Conversely, the mid-west (Dala- and Snæfellsnessýsla) and southwest coast (Gullbringu- and Kjósarsýsla) had exaggerated death counts. The corrections in cumulative death remain within 20%, though underreporting cannot be excluded.

To conclude, the Laki eruption was followed by a massive mortality crisis in Iceland, which manifested itself not in the year of the eruption (1783) but the two following years. No district escaped unscathed, but the relative severity and the timing of the
mortality peak differed between regions, with proximity to the volcano offering only limited explanation for these differences. Other, more complex processes must have been at play, including migration.

### 4.2 Mortality in parish registries

The presumed causes of death in the parish registries were grouped into 9 categories (see tab. 2 and, for detailed explanations, SM1.1). "*Landfarsótt*" (literally land-travelling disease, epidemic) and to a lesser extent "*tak*" (literally bouts of pain) were
were diseases which occurred in epidemic proportions, but it is unclear what they were and whether they were distinct diseases (some victims were diagnosed with "*landfarsótt* with *tak*"). In cases where two causes of death were given, the more proximal,





| Category | Explanation, subcategories |
|---|---|
| Hunger | includes hunger, emaciation, "misery", "swelling" (likely edema); exposure if victim poor/homeless |
| Intestinal | includes diarrhea, bloody flux, vomiting |
| Scurvy | scurvy |
| *Landfarsótt* | literally "epidemic"; some speculations about more specific use (typhus/typhoid) |
| *Tak* | literally "bouts of pain", mostly in chest; links with Bornholm disease / influenza were suggested |
| Respiratory | includes breathlessness, cough/whooping cough, chest afflictions |
| Smallpox | epidemic from Oct. 1785; distinctly recognized by pastors |
| Other illnesses | leprosy, other infectious illness, other illness, not necessarily infectious |
| Other | „sudden death", accident, old age ($\geq$ 65 years), infant (<1 year), unclear, other |
| Unspecified | Unspecified (age 1-64) |

**Table 2.** Categories for causes of death in the parish registries, as used in 4. For further explanations, see SM1.1.

lethal or specific cause was used (e.g. "leprosy and hunger" was counted as hunger, "toothache and scurvy" as scurvy, "pain and *landfarsótt*" as *landfarsótt*); otherwise, the first expression was used (e.g. in "*landfarsótt* and hunger" or "hunger and *landfarsótt*").

The presumed causes of death in parish data come with some caveats. First, reporting rules changed during the period of interest; in particular, in the north listing causes of death only became mandatory in 1785, leading to a high number of Unspecified (and hence, Old age and Infant) cases there. Second, pastors generally had no medical training but went by symptoms, possibly based on reports by family members. Illnesses with generic symptoms such as pain and fever may have been confused, whereas smallpox were widely recognised as such. Third, different pastors may have used different expressions
for the same phenomenon; for example, the expression "*innanpína*" (inner pain) only appears in the data from Útskálar parish in Gullbringusýsla. Finally, people might have suffered from more than one condition (e.g. a starving person with fever). Obvious instances of missing data (60 deaths in 3 parishes) were corrected as described in SM1.3.

Monthly mortality data for the whole of Iceland (fig. 4a) does not show a marked increase in mortality in the second half of 1783. Neither is there a significant increase of lethal respiratory disease – the fraction of deaths (fig. 4j) due to respiratory
illness was 3.2% of all recorded deaths in the second half of 1783, compared to 3.0% in the second half of 1782. That is not to say that the eruption did not induce respiratory complaints – indeed, such complaints were widely mentioned [Wieners20SM]. However, the respiratory symptoms were not (perceived to be) lethal.

Mortality in whole Iceland shows two distinct peaks, a sharp, shorter one around June 1784 and one of longer duration with a maximum in March 1785. Both peaks coincide with peaks in hunger deaths. Hunger was a seasonal phenomenon in Iceland:
Food production was highest in summer and autumn, especially in farming regions, and stores were lowest in spring.

The 1784 peak is almost entirely caused by excess mortality in the north and east; in the northeast, mortality in spring (April-June) 1784 was 24 times as high as in average 3-month period. The high proportion of unspecified causes of death can be explained by the fact that giving causes was not mandatory in the north until 1785. Unless underreporting is to blame,







**Figure 4.** Causes of death in Iceland, based on parish data. Top three rows: Relative mortality in whole Iceland and eight regions (defined in fig. 1a), broken down by cause. For whole Iceland, data is plotted at monthly resolution; for the regions, at three-monthly resolution. For example, a value of 5 in "winter 85" means that in winter (January-March) 1785, five times as many people died as normally (1773-82) would die in three months (disregarding the seasonal cycle). Note the different y-axis scales. The bottom left plot shows the fraction of death from each cause in monthly resolution. The bottom right plot shows the cumulative relative mortality from new year 1783; for example, a value of 5 in July 1785 means that 5 times as many people died in Jan 1783 - July 1785 as would normally (1773-82) die in one single year. The plain straight line denotes normal mortality (ignoring seasonality). No monthly data exist for the Southeast (Skaftafellssýslur).





disease played a relatively minor role during the first peak. In September 1784, the deputy governor wrote that in the north,
mortality was still elevated, but mostly due to disease. He also mentioned bloody diarrhoea which had infected and killed many
people during spring and summer, but which he considers to be caused by hunger [Wieners20SM]. Unfortunately, no parish
data survived from Skaftafellssýsla, the county containing the Laki craters. However, descriptive records from the parishes
closest to the volcano describe a sharp rise in mortality after New Year 1784 (Steingrímsson, 1788).

After a sharp drop in the summer months, overall mortality gradually increased again in the autumn. At first, this increase
was largely caused by an increase in disease, especially *landfarsótt*. *Landfarsótt*, *tak* and other diseases continued to contribute
significantly to the second mortality peak, while hunger and (probably hunger-related) intestinal complaints started to increase
again in autumn-winter 1784, especially in the south and southwest. Scurvy (vitamin-C deficiency) was recorded almost exclu-
sively in the coastal southwest, the mort important fishing region; fish does not contain much vitamin C and farming products
were hard to obtain in times of farming crises. In most regions, the highest mortality occurred in winter (January-March) 1785.
In the northeast and east, the 1784 peak was stronger; in the Westfjords, mortality peaked in autumn 1784 and started to decline
in winter. From October 1785, the aforementioned smallpox epidemic started in the coastal southwest and spread through the
country in the following year. As smallpox epidemics came sporadically to the distant island (Steffensen, 1972; Hálfdanarson,
1984), they tended to be severe and affected primarily people born in the period since the last epidemic had hit the country –
in this case, early 1760s (see also SM2.1).

It is difficult to determine whether hunger was also to blame for the strong increase of deaths attributed to illness in 1784-85.
Comparing *landfarsótt*, the most frequent disease, with hunger, it is clear that the two follow somewhat different dynamics.
On the national level, *landfarsótt* only seems to surge in summer 1784, after the first peak in hunger deaths, and remained
somewhat elevated after hunger subsided in summer 1785. Unless systematically underreported, *landfarsótt* seems to have
been nearly absent in the northeast in 1784, where hunger was rampant. On the other hand, in the Westfjords where hunger
was mostly absent in spring 1784 and remained comparatively mild, *landfarsótt* (and *tak*) caused a peak in mortality in autumn
1784. In addition, hunger was relatively more likely to kill persons in their prime of life, while *landfarsótt* preferably attacked
age groups that were at high risk in normal times, i.e. the elderly and the very young. Hunger was also more likely to kill
people in lower social positions – paupers and farmhands – while *landfarsótt* also killed farmers and their families (see also
SM3.1). *Landfarsótt* was thus more than a plain deficiency disease, otherwise its distribution should have resembled that of
hunger (Hálfdanarson, 1984).

This does not imply that the rise in lethal diseases 1784-85 was a coincidence. It is typical during famines that mortality
from endemic diseases increases (Mokyr and Ó Gráda, 2022), probably due to lowered physical resistance and/or disruption
in peoples' habits, e.g. because of migration. Diseases may then also infect the better-off, who would not have been in risk of
direct starvation. In other words, (endemic) disease may amplify famine mortality.
Interestingly, mortality from seemingly unrelated causes of death, such as accidents, unspecified and uninfectious diseases,
and leprosy, also strongly increased during the famine (SM3.2). Leprosy kills slowly, and mortality solely from this cause
should be independent of short-term effects such as famine. Yet recorded deaths from leprosy rose in parallel with, or even
slightly ahead of, hunger deaths. Physically weakened and socially marginalised, leprosy victims may have been among the



first to lack sufficient food, and starvation, rather than leprosy itself, may have been the proximal cause of death. It seems likely
that the famine caused more excess mortality than the nominal hunger deaths. It is plausible, though impossible to prove, that
hunger indirectly caused most of the excess mortality attributed to disease before the outbreak of the smallpox epidemic.

Any hypothesis explaining the excess mortality in 1784-85 must answer the question how an eruption lasting till February
1784 (plus potential additional factors) caused one mortality peak in spring 1784 centred in the northeast and a second one in
spring 1785 centred in the west. We will show that hunger and disease can provide a plausible explanation, but not pollution.

**5    Hunger and disease alone may explain excess mortality in Iceland**

Several factors influenced the local severity of the famine: The impact of the eruption on livestock, additional processes affect-
ing livestock, in particular the weather, the availability of alternative food sources, and migration. The loss of farming animals
can be roughly estimated by comparing reports on the number of livestock in Iceland taken in conjunction with the general
censuses of 1703 and 1785 (fig. 5a). It should be kept in mind that the 1703 reference data was obtained 80 years before the
event and was only a snapshot taken after a series of bad years. Having said this, the loss of animals seems roughly consistent
with reports of pollution. The Skaftafellssýslur, being closest to the volcano, were badly affected. From there, much of the
ash and gas seems to have been blown towards the northwest, over the uninhabited parts of Rangárvallasýsla into Árnes-,
Borgarfjarðar- Mýra- and Hnappadalssýsla (Pétursson et al., 1984), which also suffered severe losses. In those counties, the
upland farms typically were hit worse than the coastal regions, probably due to the direction of wind dispersal. Kjósar- and
Gullbringusýsla further south escaped rather lightly, as did the Westfjords (Barðarstrandar- and Ísafjarðarsýsla) further north
- the only region where no ash layer was reported. Coastal regions also profited from access to seaweed as additional fodder.
The losses in the North were around average. The Múlasýslur seem to have escaped lightly, but the losses there may have been
underestimated due to the low 1703 reference value.

Fig. 5b shows that human mortality and loss of animals were only weakly correlated across counties (correlation 0.41 when
using relative mortality, and 0.21 using death rates). Using the "corrected" baseline for 1703 animal ownership (defined in
3.2) instead of the original 1703 values does not lead to a stronger correlation. The Westfjords escaped lightly both on human
mortality and on animal loss. In the north, animal loss was around the national average, yet human mortality was far higher;
in Gullbringusýsla, despite mild loss of livestock, human mortality was above average. In Árnes- to Skaftafellssýsla, human
mortality was low compared to the loss of livestock, though in Skaftafellssýsla outmigration helped to keep human death counts
low. In the west, the picture is mixed: In Borgarfjarðar- and Dalasýsla, human mortality was low compared to livestock loss,
while for Mýra- and Snæfellsnessýsla, the outcome depends on which mortality measure is used.

By far the most significant non-livestock source of food was fish (fig 1b). The main fishing regions were in the west and
southwest. In the north and east, many lived in inland farms and those who lived near the shore could only fish during summer,
which was also hay harvest season. In the summer of 1783, attempts to fish largely failed due to hazy weather. During the winter
and spring 1783-84, results were mixed in the southwest and west, but on the whole not abnormally bad [Wieners20SM]; the
Danish merchants were able to buy the usual amount of dried fish. In the north, fishing in spring was hampered by sea ice,





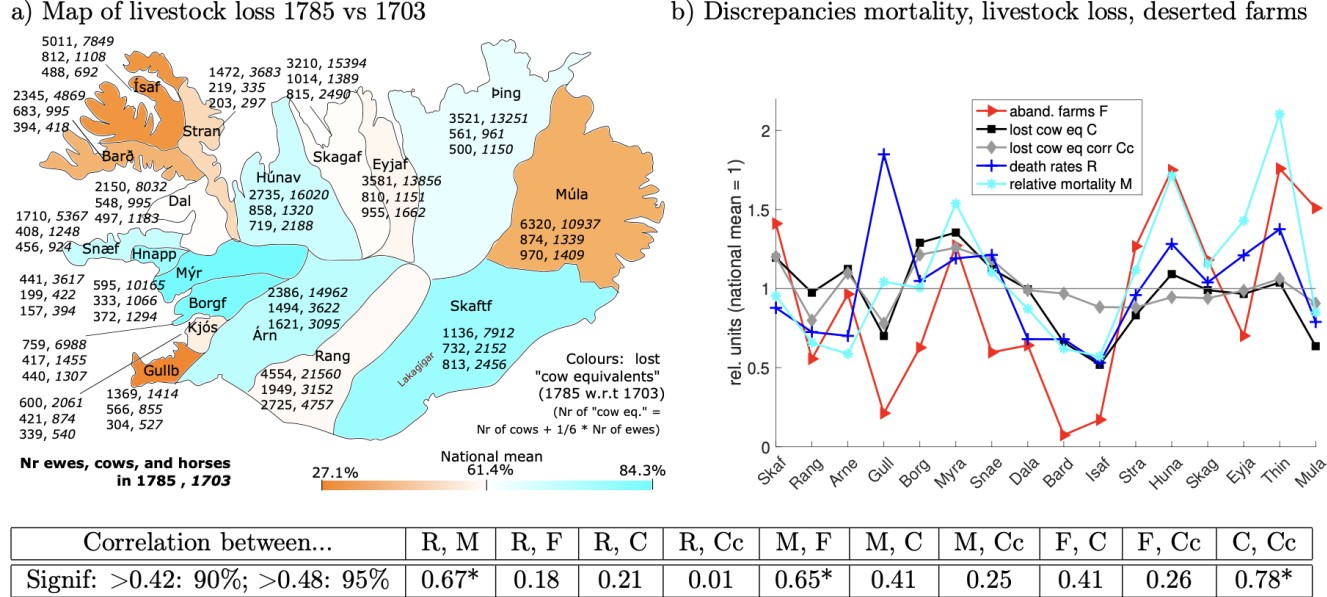

| Correlation between... | R, M | R, F | R, C | R, Cc | M, F | M, C | M, Cc | F, C | F, Cc | C, Cc |
|---|---|---|---|---|---|---|---|---|---|---|
| Signif: >0.42: 90%; >0.48: 95% | 0.67* | 0.18 | 0.21 | 0.01 | 0.65* | 0.41 | 0.25 | 0.41 | 0.26 | 0.78* |

**Figure 5.** a) Animal loss 1785 w.r.t 1703. The colour code depicts the (uncorrected) fraction of cow equivalents (E) lost, i.e. (E(1703 – E(1785))/E(1703). Six ewes count as equivalent to one cow. The colour code is tuned such that the darkest orange (blue) correspond to the least (worst) hit area, and white to the (weighted) mean for whole Iceland. The numbers for each county depict the number of ewes (top), cows (middle) and horses (bottom) for 1785 (left) and 1703 (right, italics). The data is also presented in table form in SM2. b) Relative mortality M, death rates R, animal loss in cow equivalents (plain C and corrected Cc, see sect. 3.2) and abandoned farms per region F, normalised by the weighted mean for whole Iceland. The correlations between the quantities in this plot are given in the table; correlations are significant at 90 (95) % confidence if their absolute value exceeds 0.42 (0.48), hence correlations marked by a (*) are siginficant at 95% confidence and others not even at 90%.

and the worst-hit region, the northern part of Þingeyjarsýsla, had no catches even in summer 1784 [Wieners20SM]. In normal years, farmers from the north often travelled to the main fishing regions to work there or barter fish for farming products, but in 1784, they lacked produce, horses, and healthy men for the journey.

An additional source of foodstuff was aid from Danish trade posts. In general, the density of trade posts was highest in the fishing districts, while the southern coast was a harbourless stretch of about 400km. The aid obtainable at trade posts depended on the willingness of the merchant, the strong-mindedness of the district commissioners in coaxing the merchants to help, and the availability of foodstuff in the store houses (Wieners, 2020). These factors varied strongly. For example, Hálfdanarson (1984) suggested that the death toll in Húnavatnssýsla 1785 may have been exacerbated by the fact that no trade

made it to the local trade posts that summer due to sea ice. Conversely, merchants in Eyrarbakki and the Vestmannaeyjar in the south provided aid to the fugitives from Vestur-Skaftafellssýsla (Steingrímsson, 1788). Overall, famine relief provided through the Danish government via the trade company was meagre, amounting to about 2 weeks of additional provisions for the whole of Iceland in 1784 (Wieners, 2020) and arrived only from summer 1784 onwards – too late to mitigate the very high




mortality in spring 1784 in the northeast. As opposed to others in need, farmers and fugitives from near the eruption site also
received monetary aid from the Danish government (Gunnlaugsson, 1984a, b; Jónsson and Sigurðsson, 1784; Finnsson, 1785; Skúlason, 1784), probably not least because the Danish government was more interested in relieving victims of the spectacular destructions wrought by the eruption and keeping farms functional than in preventing starvation (Einarsson, 2022).

To understand why mortality in the northeast peaked almost a year earlier than in the southwest, one needs to consider the weather in the years before and after the eruption. The northeast experienced a harsh winter 1781-82, poor grass growth in 1782,
and again a harsh winter 1782-83 (Guðjónsson, 2010, Wieners20SM). The ill weather led to loss of animals in the northeast even before the eruption. In Þingeyjarsýsla, reportedly only three cows were left in the northern peninsula of Melrakkaslétta, and similarly few in the neighbouring Langanes (Guðjónsson, 2010). In mid-April 1783, the county commissioner of Þingeyjarsýsla wrote that in the northern half of the county, many farms were without a single horse, cow or sheep, and while not many people had died yet, many were bed-ridden from weakness, unable to work, or had left their farms and would become
a burthen to the better-off southern half of the county (Jónsson, 1783). Mortality in Þingeyjarsýsla in 1783 was about 60% above the 1773-82 average (fig. 2a), suggesting a relatively minor local famine. Neighbouring Eyjafjarðarsýsla also had some loss of animals in 1782-83 [Wieners20SM] but no excess mortality. The early reduction in livestock likely led to a depletion of reserves - from food stores to human strength and body fat - prior to the eruption. This may help explain why Eyjafjarðar- and especially Þingeyjarsýsla were hit not just badly but also early. To exacerbate the food shortage, milk yields in cows and ewes
dropped dramatically after the eruption began. In the northeast, the drop in yield per animal thus coincided with an already low number of animals.

In contrast, the southwest and west, from Rangárvalla- to Snæfellsnessýsla, had only low excess mortality in the first half of 1784. These regions were much less affected by cold weather prior to the eruption and may have had food reserves to last most households through the first winter. However, the southwest, especially Rangárvalla- and Árnessýsla, had a wet summer
in 1784, which greatly hindered the hey harvest [Wieners20SM]. It seems therefore possible that part (but only part) of the livestock loss in this region only occurred a year after the eruption. By the time the number of animals reached its minimum, the milk yield per animal may have recovered. Also, some, albeit modest, help from Denmark arrived in summer 1784, and the inhabitants of Rangárvalla- and Árnessýsla were able to sell some of the animals they could not feed to farmers from the north and east (Gunnlaugsson and Rafnsson, 1984).

Migration was generally directed westwards, especially towards the fishing regions. Snæfellsnessýsla received a high amount of fugitives from the north, and Gullbringusýsla from the south, especially Vestur-Skaftafellssýsla, plus a smaller number from the north. Migration over shorter distances, including within counties, also occurred and was often directed coastwards. It modulated mortality in several ways. First, it affected the death counts, as dying migrants were registered in the location of burial, not the places they had fled from. Second, migration was directed towards the areas of relatively high food supply
(fish), therefore it probably mitigated shortage in the worst-hit areas and saved the lives of some migrants who might otherwise have died. The inhabitants of the parish closest to the volcano may have been "lucky" in the sense that the destruction of the livestock manifested itself already in summer-autumn 1783, giving the people time to flee before winter arrived. On the other hand, immigration increased food shortage in the receiving regions, especially if the migrants were too weakened to work. The





county commissioner of Snæfellsnessýsla complained in 1784 that hundreds of weakened fugitives from the north formed "no
small burden to this county" while a lack of able-bodied seasonal fishermen from the northern counties prevented 50 boats from
being manned [Wieners20SM]. Finally, migration might have helped the spread of infectious diseases, by causing people from
different regions to mix, making weakened homeless persons walk from house to house in the search of shelter, and leading to
overcrowded accommodation, for example fishing huts. However, the spread of diseases was probably also dictated by chance.

To summarise, the following storyline emerges, which can explain regional discrepancies in terms of hunger and disease:
Close to the volcano, livestock loss was large and rapid, but many people were able to flee, keeping local death count lower;
in the northeast, prior loss of livestock, lack of alternative food sources and insufficient opportunity to flee made sure that
the people were hit early and disproportionally hard; the Westfjords escaped lightly due to low exposure; in the West and
Southwest, most people had enough reserves, and in coastal regions access to fishing, to survive the first winter, but bad
weather and immigration helped to deplete food availability enough to cause hunger in the second winter, while diseases
(exacerbated by immigration and overcrowding) contributed strongly to mortality.

## 6 Mortality outside Iceland

### 6.1 Mortality data outside Iceland - review

We did not analyse primary mortality data for regions outside Iceland, but briefly review existing literature.

For England, Grattan et al. (2003) find that mortality in England in July-September 1783 was 29% above its 51-year truncated
running mean[2]. The excess mortality was negligible in July (15000 deaths in 1783 vs 14747 in the running mean), became
notable in August (18338 vs 14429) and peaked in September (22751 vs 14372).

Witham and Oppenheimer (2004) estimate an excess mortality of 11500 deaths for the whole of England in August and
September 1783 (40% above the 1759-1808 mean), and an additional 8200 during a second mortality peak in January and
February 1784 (23% above the mean). Overall, the east of England was hit hardest, but the spatial distribution of "crisis
mortality" (Z-score >2) showed much small-scale variation, with parishes that did / did not experience crisis mortality often
side by side. Also, counties which experienced crisis mortality in summer 1783 did not necessarily show crisis mortality in the
following winter, and vice versa.

Using a larger set of parish data, Hellman (2021) finds that in England, the number of deaths in 1783 and 1784 were not
extraordinarily high (1783: 58943, 1784: 56198; mean 1770-99: 58298; standard deviation 1770-99: 3202). Looking at monthly
data, his results qualitatively confirm those of Grattan et al. (2003) and Witham and Oppenheimer (2004) as he, too, finds a
mortality peak in August-October 1783 and a second one in early 1784. Unfortunately, a direct comparison is impossible due
to discrepancies in reporting methods, in particular the lack of seasonal cycle data for normal years in (Hellman, 2021). In
Wales, Hellman (2021) finds 1783 and 1784 to have the highest mortality in 1770-99 (1783: 2241, 1784: 2474; mean: 1814,
std: 231), without giving seasonal data.

---

[2]Grattan et al. (2003) gives a higher increase, 69%, but table 3 of that paper seems to have miscalculated the running mean for the whole summer, giving
33159 instead of 43548, which equals the sum over the individual months and looks largely consistent with their fig. 1.





For France, Balkanski et al. (2018) report excess mortality during summer (June-September 1783) in the parishes with available data: 4093 persons died in that period[3], compared to 3092 in the 1774-1789 mean, an increase by 32%. Grattan et al. (2005) found that mortality was elevated by 38% in August-October for a smaller set of French parishes. If representative for the whole country, this would amount to about 16000 excess deaths. Hellman (2021) reports no unusual mortality on annual level (1783: 988, 1784: 927; mean 1770-99: 882, std: 102).

In a small dataset from the Netherlands, Hellman (2021) finds that 1783 had the third-highest morality in 1770-99 (1783: 2138, 1784: 1682, mean: 1719, std: 330). On a monthly level, a larger peak in late summer-autumn 1783 (max in September) and a secondary peak in January 1784 are observed. For Sweden and Norway, Hellman (2021) finds no extraordinary mortality on annual level, though Sweden shows a mortality increase in November 1783-February 1784. For Norway the signal is less clear. Lack of background seasonal cycle data prevents any estimate of excess mortality for these months.

To summarise, the available data for Central-Western Europe suggest some excess mortality peaking in August-October and again in early 1784, though in most cases the signal was too weak to show in the annual data. In Sweden, absolute mortality increased only in November, but increases relative to the usual seasonal cycle could not be determined.

## 6.2 The role of hunger and disease outside Iceland

Despite wide-spread damages to vegetation (Grattan et al., 2003), historical records show no evidence for harvest failure in 460 1783 in west and central Europe (Halldórsson, 2013; Witham and Oppenheimer, 2004). So probably the vegetation damage, while dramatic in its sudden and unusual appearance, was not sufficiently large to cause massive harvest failure over most of Europe. Even if harvest failure had occurred, it would probably not have caused hunger already in summer and autumn 1783. Balkanski et al. (2018) point out that in France the mortality did not affect the poor people more severely than the wealthy, as would be expected if hunger had been a main cause.

Witham and Oppenheimer (2004) discuss the possibility that the warm summer 1783 may have caused at least part of the excess mortality in England from August onward. July 1783 was the warmest July in their data set, and they point out that hot summers tended to cause excess mortality after a few weeks, probably because the heat stimulated the spread of disease, for example by fostering insects which could transmit disease. Anecdotal evidence (Grattan et al., 2003; Hellman, 2021) likewise points at the occurence of fever. The second mortality peak in England, in early 1784, can at least partly be explained by cold, 470 potentially aggravated by disease (for example, people huddling close together for warmth might spread typhus-infested lice). However, Witham and Oppenheimer (2004) find that the warm July 1783 and the cold winter 1784 only explain about 1/3 and 1/2 of the corresponding mortality peaks, which may be due to missing additional factors but also to limitations of their linear regression method.

One interesting aspect about the English mortality data is the fine-grained variability in space (Witham and Oppenheimer, 475 2004), with strongly and weakly affected parishes in close proximity. This might be consistent with disease outbreaks, depending on how much people travelled among parishes and how easily diseases were transmitted. Local geography might also

---

[3]They made a minor typo, writing 4193 instead of 4093





have played a role, for example the proximity to some insect-infested pond or swamp, or regional climate effects (e.g., altitude reducing summer temperature).

To summarise, contagious diseases triggered by climate anomalies may explain at least a substantial part of the western
European excess mortality, but it is unclear whether they can explain all of it.

## 7   Human death by haze?

The hypothesis that volcanic air pollution directly contributed to human mortality seems to rest on the fact that about a month after the haze and associated respiratory symptoms were observed in Europe, significant excess mortality occurred. However, this does not prove that the concentration of harmful substances, particularly SO2 and PM2.5 (particulate matter of less than
$2.5\mu m$ diameter) really were a significant cause of death in 1783.

### 7.1   Morbidity vs Mortality

There is ample evidence for haze-induced respiratory symptoms both in Iceland and Europe. In Iceland, the *þingvitni* (farmers' statements) in Eyjafjarðarsýsla, Dec. 1783, reported 'disgusting stench and ill odour, such that men with [pre-existing] breast diseases temporarily stayed in bed' [Wieners20SM], while Ólafur Stephensen in west Iceland wrote that 'The sulphur stench is
so strong, that people cannot draw their breath well, and it is not unreasonable, that this can cause disease' (cited in Thordarson (1995)). Close to the volcano, Reverend Jón Steingrímsson mentions respiratory disorders such as difficult breathing, especially with persons suffering from pre-existing chest diseases, and irritated throats, skin and eyes (Steingrímsson, 1788, p. 41). These symptoms persisted at least through spring 1784, possibly due to continuing outgassing from the lava streams. None of these sources mention human deaths directly connected to these symptoms, or a mysterious increase in mortality in summer-autumn
495  1783.

In Europe, Grattan and co-workers showed that health problems consistent with health volcanic pollution were wide-spread, see e.g. Durand and Grattan (1999, and references therein). Some contemporary sources in Europe link the haze to illness and mortality Grattan et al. (2003, and references therein), although as noted above, in some cases the wording suggests contagious disease rather than pollution.

Evidence for morbidity induced by long-term exposure to volcanic SO2 and sulphate aerosol is also documented for modern eruptions, see Stewart et al. (2022) for a review. Their table 2 lists various cardio-pulmonary symptoms in exposed persons, but the reviewed studies do not document severe illness or excess mortality. Probably this is at least in part because the studied populations were not large enough - typically a few hundred - to support mortality statistics.

By contrast Wakisaka et al. (1988) found long-term (11 year) exposure to SO2 and aerosol from Sakurajima volcano to be
associated with increased mortality rates due to non-infectious respiratory disease (14% risk increase in the area up to 30km from the volcano), though only bronchitis showed a consistent relationship with both distance from the volcano and temporal variation of concentrations. However, the effect was not visible in *total* mortality rates.



Therefore, while haze from the Laki eruption clearly induced morbidity, this does not necessarily imply significant effects on (total) mortality.

## 7.2  The time lag between haze occurrence and mortality

As noted by Witham and Oppenheimer (2004), Grattan et al. (2003, 2005), and Hellman (2021), excess mortality in England and France only rose in August 1783, while strong haze had been present from the end of June (Thordarson and Self, 2003).

This is in contrast with modern pollution events, where faster responses were observed. For example, Micheaud (2004) found that hospital emergency room visits for asthma and Chronic Obstructive Pulmonary Disease (COPD) followed exposure to SO2 and fine sulphuric acid aerosol at Kilauea, Hawaii, at lags of 1-3 days. Similarly, Carlsen et al. (2021a, b) studied the effect of chemically young (SO2-rich) and chemically mature (aerosol-rich) plumes from the 2014 Bárðarbunga eruption in Reykjavík and found significantly increased demand for healthcare at 0-2 days lag for both types of plumes.

Such findings do not exclude health effects at longer lags - these would be hard to discern or disprove with data from fluctuating plumes. More prolonged health effects probably occurred after a major non-volcanic pollution event, the London smog of December 1952 (Bell and Davies, 2001). The smog, which lasted 5 days, was accompanied by a sharp mortality peak. In the 9 days around the smog episode, mortality was strongly correlated to pollutant (SO2 and particulate matter) concentrations at 0-1 day lags, but mortality remained elevated for several months following the smog, suggesting a long-term effect to pollution. However, this long-term effect followed a stronger immediate impact.

If these modern analogues are representative of historical haze-induced mortality, it is puzzling that mortality in England and France should only show a long-term response without an immediate impact. In Iceland, the lag between the strongest haze and the onset of the mortality crisis (December 1783) is even longer. As explained above, overall mortality in 1783 was below average in Iceland. In addition, the seasonal distribution of deaths in Iceland is close to normal non-famine years (fig. 6). In particular, the fraction of deaths occurring in July and August is slightly below normal, whereas the fraction in December is elevated - the first increase leading to the mortality peak in spring 1784.

Apart from timing, the strong differences in mortality between nearby parishes in England (Witham and Oppenheimer, 2004) does not seem to fit with Laki pollution as the main cause. The polluted air was transported to England at great height before descending in a high pressure area (Thordarson and Self, 2003), which should result in relatively homogenous concentrations of pollutants and hence mortality over scales of a few hundred kilometers - unless local variation in susceptibility (e.g. urban vs rural populations) and small-scale geographic effects such as sheltering from prevailing winds played a major role. However, studying the latter would require simulating pollution at much higher resolution (order of 1km) than hitherto attempted (order of 100km in (Balkanski et al., 2018)).

## 7.3  Modelling studies of the haze

Several modelling studies simulate concentrations of SO2 and PM2.5 after the Laki eruption. In table 3, their findings are compared to health guidelines and measurements from modern pollution events.





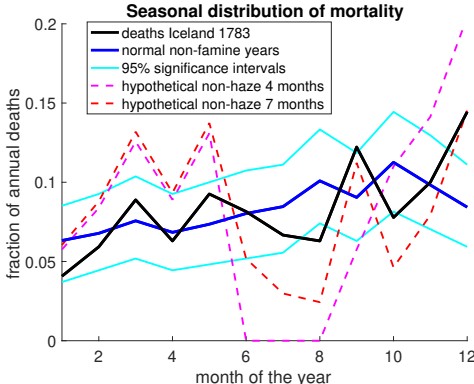

**Figure 6.** Seasonal distribution of deaths in Iceland. For any period, the plot shows fraction of deaths occurring in a given calendar month, based on the parish registers. Data is shown for the $N = 270$ deaths in 1783 covered in our parish data, and for the mean over four famine-free decades (1740-49, 1760-69, 1770-79, 1790-99) (Vasey, 1991). 95% confidence intervals were computed by creating 10000 randomised surrogate distributions and taking the 5th and 95th percentiles for each month. Each surrogate distribution was generated by randomly assigning a month of death to $N = 270$ persons using the probability distribution of the famine-free years. The actual 1783 distribution is thus within normal fluctuations if it is inside the 95% significance interval. The two "hypothetical" scenarios assume that 400 out of 1227, i.e. out of the total number of deaths in 1783, can be attributed to haze, in line with (Balkanski et al., 2018). These haze-related deaths are evenly spread over 4 months (June-September) or 7 months (June-December). The graph shows the seasonal distribution of the remaining 827 deaths. A slightly negative value for July in the 4-months curve was set to zero.

540   Reporting criteria vary strongly across modelling studies, but collectively they suggest that health guidelines for SO2 and PM2.5 were at most mildly transgressed in Europe. The reported concentrations are also much lower than the concentrations measured around Sakurajima/Japan (table 3c), which caused no significant excess all-cause mortality despite much longer exposure time.

   For Iceland, modelled concentrations differ widely. Balkanski et al. (2018) find values somewhat lower than the observations
545   from Sakurajima, while the PM2.5 concentrations found by Schmidt et al. (2011) and the (probably overestimated) values of Chenet et al. (2005) are significantly higher.

   Balkanski et al. (2018) and Schmidt et al. (2011) also model the mortality effect of the Laki eruption itself or a present-day Laki-like eruption, respectively. This requires assumptions on the "response functions" of excess mortality to volcanic haze. As pointed out in these studies, response functions suffer from high uncertainty. Observations are scarce, in particular on
550   intermediate time scales of a few months, as typical pollution events are either short-lived or chronic exposure; pollution events tend to contain a mixture of pollutants whose effects are hard to disentangle; response functions are based on generic particular matter and ignore that acidic aerosol may have stronger effects. Estimating response functions for pre-industrial times suffers additionally uncertainties including different background exposure (less industrial pollution but likely more smokey houses), a different age profile, and a less developed healthcare system.



Balkanski et al. (2018) modelled increased risk of death for parishes in France where observations exist and found relative-risk values of 0.4-3% for June-September 1783, far below the observed excess mortality (32% above average). They conclude that the haze alone cannot explain observations. Schmidt et al. (2011) find ≈140000 additional deaths in Europe for a Laki-type eruption in modern settings. For England, they report 21000 deaths, which amounts to 3-6 times lower excess mortality rates than observed in 1783 (Grattan et al., 2003), though direct comparison is problematic seeing that they assume modern background pollution. Also, they include only the effect from PM2.5, not from SO2, although this might be acceptable at least for Europe where plumes were likely to be chemically mature). A follow-up study (Heaviside et al., 2021) finds 772 deaths for the UK as best estimate (3350 as worst-case scenario), i.e. even lower numbers.

For Iceland, Balkanski et al. (2018) find an increased risk of dying of 13-30% from SO2 and 60% from PM2.5 for June-September 1783. Assuming the effects of SO2 and PM2.5 to be additive, this implies a risk increase by $\approx 80\%$ (73-90%). For an average annual 1500 deaths in Iceland (mean 1750-1800) and a 80% increase in mortality rates over 4 (6) months implies about 400 (600) excess deaths. These numbers are small compared to the excess mortality following the eruption (≈8000 deaths in 1784-85), but might suggest a modest contribution for 1783. Schmidt et al. (2011) find an excess mortality of 400 deaths for present-day Iceland, which implies lower death rates than Balkanski et al. (2018), given the roughly 6 times larger population on 2004 compared to 1783. The discrepancy may partly be due to the omission of the effect of SO2 in Schmidt et al. (2011).

No excess mortality was actually observed for Iceland as a whole in 1783. However, as mortality in Iceland fluctuated strongly (standard deviation 1750-1800: 937 deaths/year), it cannot be excluded that without the eruption, mortality in 1783 would have been below-average, masking a possible impact of the haze. But assuming that 400 deaths out of 1227 observed in 1783 were caused by haze, as suggested by Balkanski, would leave only 827 deaths by other causes, which would imply a lower death rate than ever observed 1750-1800 (16 deaths / 1000 inhabitants; actual minimum: 18.5) and a highly atypical seasonal pattern (fig. 6). Assuming a smaller haze-induced mortality of 200 deaths over 7 months would still imply an unusually low non-haze relative summer mortality (not shown).

If Icelandic pollution-induced excess mortality in 1783 likely stayed below 200 (about 26% of the normal death toll for 1/2 year), it seems unlikely that Northwestern Europe experienced an excess mortality of that order of magnitude, even though pollutant concentrations there were much lower than in Iceland.

To summarise, while there is clear evidence for haze-related morbidity in Iceland and Europe, this does not hold for mortality. In Europe, observed excess mortality was much stronger and followed a different timing than (imperfect) modern analogues suggest if haze had been the main cause. In Iceland, there is no evidence for any haze-related excess mortality in 1783. Of course, this does not exclude that some (up to $\approx 200$) individuals did die from pollution. It is also conceivable that haze-induced morbidity contributed indirectly to excess mortality in Iceland and/or Europe, for example by weakening people's resistance and making them more vulnerable for hunger and disease. In fact, pollution from forest fires has been linked with increase risk of influenza months later (Landguth et al., 2020). However, evidence for direct haze-induced excess mortality in Iceland or Europe remains weak. In particular, air pollution does not seem to explain the large death toll in Iceland during 1784-85.





a) recommended max. concentrations in $\mu g/m^3$ according to IVHHN (2022). The three values show the strictest guideline; EU guideline; least strict guideline cited.

| SO2, annual | SO2, daily | PM2.5 (PM10), annual | PM2.5 (PM10), daily |
|---|---|---|---|
| 20; 20; 101 | 51; 125 (max 3 days/year); 407 | 15; / ;15 (20; 50; 150) | 65; /; 65 (50; 50 (max 35days/year); 250) |

b) Modelled concentrations during the Laki haze (in $\mu g/m^3$).

| study | SO2 Iceland | PM2.5 Iceland | SO2 Europe | PM2.5 Europe |
|---|---|---|---|---|
| Chenet et al. (2005) (*) | | up to 600 | | up to 60 Northwest Europe |
| Oman et al. (2006b), mean June-August 1783 (**) | 25-50 | 8-20 (***) | | |
| Schmidt et al. (2011) (****) | | South (rest of) Iceland: > 100 (50-100) | | England (France): 20-30 (10-20) |
| Balkanski et al. (2018) (June-Sept 1783) | 125 exceeded 55 days (highest daily: 294 ) | 20 exceeded 70 days; mean (highest daily-mean) value 34 (148) | 125 exceeded on 0 days in France | mean (highest daily-mean) value in France: 3.1 (71); 20 exceeded up to 3 days in France |

(*) assuming 100% conversion of SO2 to sulphate aerosol, hence aerosol likely overestimated. First 2 months after eruption.

(**) Average over wide area, $30°W - 45°E$ along $65°N$ . Local concentrations in Iceland likely higher.

(***) Including all sulphate aerosol, not just PM2.5. Probably only mild overestimate, see remark table c.

(****) *increase* due to Laki-style eruption relative to present-day conditions, i.e. influenced by industrial activity.

c) Concentrations (in $\mu g/m^3$) during historical pollution events with mortality effects

| Event | SO2 | Part. matter | Impact of exposure |
|---|---|---|---|
| Sakurajima/Japan Wakisaka et al. (1988). overall mean; highest annual; highest daily (**) | 38; 122; 434 | Total suspended matter (*) 29; 54; 155 | Risk of death by non-infectious respiratory disease 14% increased 0-30km from volcano. Out of 3 subcategories, only bronchitis shows consistent response. No effect on total mortality. |
| London smog 1952 (Bell and Davies, 2001) 5-day mean (highest daily) | 1516 (1835) | Total suspended matter (*) 1400 (1620) | Significant excess mortality, sharp peak during smog event, persistent elevated mortality during following months. |

(*) PM2.5 is contained in Total Suspended Matter (TSM), thus PM2.5 <= TSM. IVHHN: Volcanic aerosol mostly in PM2.5 range, thus PM2.5≈TSM. For industrial smog/haze, Bell and Davies (2001) assumed 60% of TSM to be PM10. Cho et al. (2013) suggests about 90% of PM10 being PM2.5 and 80% PM1.

(**) out of 4 measurement sites providing up to 11 years of data.

**Table 3.** a) Annual and daily mean concentrations not to be exceeded, according to the International Volcanic Health Hazard Network (IVHHN). b) Modelled concentrations for Laki haze. c) Observed concentrations during haze events with recorded impact on mortality. All concentrations in $\mu g/m^3$. PMX refers to particulate matter (i.e. aerosol) with diameter smaller than X $\mu m$





## 8 Human death by fluorine poisoning?

As mentioned, $\approx 450 mg/m^2$ of fluorine fell over most of Iceland. Assuming a hay crop of $400 kg/ha = 0.04 kg/m^2$ (Friðriksson, 1972), this amounts to up to $11250 mg/kg(hay)$ (which is an upper estimate because fluorine concentrations may have been reduced in between ash showers through dilution by rain water or permanent adsorption to the soil (Thordarson, 2011; D'Alessandro, 2006)). Assuming that a sheep eats the equivalent of 3kg of dried grass/day (Sigurdarson and Pálsson, 1957) and weighs $50 kg$, fluorine intake may have been well above the dose of $15 mg/day/kg$(bodyweight) which may kill after 6 months (see sect. 8.2).

There is no doubt that grazing animals died of fluorine poisoning after the Laki eruption (Pétursson et al., 1984). Symptoms described in contemporary records (Steingrímsson, 1788; Finnsson, 1796), such as feebleness, softened bones, growths (exosthoses) on the bones, and death, match perfectly with later findings and experiments (Roholm, 1937, henceforth referred to as RoX, with X being a chapter number).

While there have been several cases of (lethal) fluorosis among livestock after ash-producing eruptions (Pétursson et al., 1984; Roholm, 1937), fluorine-related morbidity in humans mostly arises due to persistent volcanic activity causing contamination of drinking water by rock-water interaction or ashfall (D'Alessandro, 2006; Walser, 2020). This mostly concerns only dental fluorosis, i.e. stains on the enamel of teeth (Stewart et al., 2022), although fluorine concentrations sufficient to cause skeletal fluorosis may occur in some volcanic areas (D'Alessandro, 2006; Walser, 2020). Fluorosis from (non-permanent) eruptions seems to be rare (Stewart et al., 2022), the Laki eruption being the only example where such an effect has been suggested (D'Alessandro, 2006). In this chapter we first review historical and archeological records, and then provide rough estimates of potential fluorine uptake by humans.

### 8.1 Historical and archeological records

Many contemporaries considered it possible that humans and animals suffered from the same disease, described by some as "bone sickness" or "a kind of scurvy" (Stephensen, 1785, Wieners20SM). Parish registers do not list "bone sickness" as a cause of death. "Bone pain" occurs 15 times, mostly in connection with other symptoms (*tak*, headache, weakness, old age). None of these cases occurred in 1783, five in 1784 (three of those in the first six months), six in 1786 and 4 in 1786-87. This does not suggest a surge of fatal "bone sickness" in the months following the eruption, although cases in 1783-1784 may be slightly underrepresented due to higher proportions of unspecified deaths.

Parish registries do list 99 deaths from scurvy. The fact that farming animals were wrongly diagnosed as having died from scurvy (obviously a misdiagnosis, as sheep, cows and horses can produce vitamin C) may at first imply that human victims of fluorosis, if any, could have been misdiagnosed as scurvy victims. Of the 99 diagnosed scurvy cases, 88 occurred in Gullbringusýsla, mostly in the first half of 1785. These deaths thus occurred well after the fluorine exposure peaked, but coincided with the peak in hunger deaths. Furthermore, Gullbringusýsla had the lowest animal loss, suggesting a relatively modest exposure to fluorine, whereas the local diet was high in fish and thus low in vitamin C. The recorded cases are thus probably due to genuine scurvy caused by the fish-based and hence vitamin-C-deficient diet, rather than misdiagnosed fluorosis.





The rapid loss of livestock indicates that parishes closest to the eruption experienced the highest exposure to fluorine. As mentioned in the introduction, observations of the local pastor (Steingrímsson, 1788) and a young emissary of the Danish government (Stephensen, 1785) have been interpreted as symptoms of fluorosis in humans.

However, this assessment appears somewhat superficial, as most symptoms in these sources match hunger and scurvy better than fluorosis (table 4). In particular, black swollen gums and loss of teeth are signature symptoms of scurvy. "Swellings and bloated bodies", "Ridges and growth", and inability of patients to stretch their legs could hint at deformation of joints or bones through exostoses (material accretion on the bones) caused by fluorosis. However, exostoses require high fluorine intakes of about $10mg/day/kg$ bodyweight (RoXXII), far exceeding our estimates of human fluorine ingestion (sect. 8.2). In addition, these symptoms can be well explained by scurvy, possibly combined with hunger edema. The fact that the above symptoms improved upon consuming food containing vitamin C also points to scurvy as the main cause. The only reported symptom that fits better with (strong) fluorine poisoning is thirst and excessive urination.

Perhaps more conspicuous is the absence of two classic symptoms of chronic fluorosis from table 4. First, skeletal fluorosis in humans leads to stiffness in the spine because of calcification of the ligaments; this key symptom was found by Roholm in cryolite workers (RoXIV), but is not mentioned in Iceland after 1783. If fluorosis had been severe enough to cause "ridges and growths" through exostoses, one would expect spinal stiffness to be observed as well. Second, prolonged exposure to fluorine, even at low doses, during childhood causes dental fluorosis (RoXIX). Bishop Hannes Finnsson carefully studied the phenomenon in animals in the years following the Laki eruption (Finnsson, 1796), but does not mention similar symptoms in humans. This suggests that even dental fluorosis was not widespread.

Archaeological investigations also do not provide evidence of lethal fluorine poisoning. Gestsdóttir et al. (2006); Walser (2020) analysed human skelettal remains, including from the period of the Laki eruption, but found no signs of clinical fluorosis.

All in all, written and aerchaeological records do not support the hypothesis of (wide-spread) lethal fluorosis in humans. One cannot exclude, however, that elevated fluorine intake exacerbated scurvy, because fluorine might impede the utilisation of vitamin C in the human body (RoXXVII).

## 8.2 Estimates of fluorine ingestion

As a complementary line of evidence, we provide an upper estimate of human fluorine ingestion and a low estimate of the critical dose for morbitity and mortality. This will allow to assess whether widespread illness and death by fluorine poisoning was at least plausible.

For a lower bound of the lethal dose of long-term fluorine intake, we use RoXXI-XXIV. The author induced chronic fluorosis in rats, pigs, calves and dogs. Feeding these animals roughly $15mg/day/kg$(bodyweight) of fluorine caused strong symptoms but was in most cases not lethal within 1/2 year. One pig died on day 171, some rats after 0.6 to 1.5 years. So $15mg/day/kg$ may serve as a lower bound for the lethal dose of fluorine for 1/2 year. For a human (or sheep) weighing 50kg, this equals $750mg/day$.

To obtain a lower estimate of fluorine intake causing symptoms other than dental fluorosis (stained teeth), we note that Prashuta et al. (2011) found that $10mg/day$ ingested for 10 years or more can cause skeletal fluorosis (bone changes), and



Human symptoms 1783-84 in historical sources and possible explanations in terms of fluorine poisoning, hunger and scruvy

| symptom | fluorine poisoning | hunger | scurvy |
|---|---|---|---|
| swellings, bloated body (JS, also MS) | "thickening of extremities" in animals (unclear, may be bone thickening?) | hunger edema | yes |
| ridges and growths on ribs, back of hands, legs, feet, joints (JS, also MS) | knots on bones typical, but seem to need high dose, $\approx 10mg/day/kg$ bodyweight (RoXXII) | edema? | swollen, inflamed joints, swollen legs |
| contracted sinews, unable to stretch leg (JS, MS) | no (unless joint deformation was the cause) | no | Considered key symptom (and origin of Icelandic name *kreppusótt*) by Steffensen (1972). Pseudoparalysis / relief posture due to pain in bones / inflamed joints ? |
| bloating, swollen joints, knotted sinews quickly improve upon eating dandelion broth (JS) | no | no | dandelion provides vitamin C. Scurvy can improve within days of starting vitamin C treatment. |
| black swollen gums, loss of teeth (JS, MS) | no (other types of tooth deformation can occur) | no | classic symptoms |
| gum swellings improve upon getting fresh milk (JS) | while milk may help against fluorosis (calcium, vitamins D and C), gum swelling is no fluorosis symptom | no | milk contains vitamin C |
| sore growth on neck and tighs (JS) | no | no | subcutaneous bleeding, bad healing of wounds and reopening of old wounds, in later stages bleeding |
| rapid heartbeat, lack of breath (JS) | no | rapid heartbeat yes; lack of breath possible (e.g. due to anemia) | yes |
| loss of hair (JS) | maybe (JS: some loss of hair in poisoned animals; RoXXI-XXIV: only "untidy" or "coarse" coat) | yes | no |
| thirst (OF), excessive urination, incontinence (JS) | yes (in animals at high dose, RoXXI-XXIV | not a typical symptom | no |

**Table 4.** Human symptoms recorded after the Laki eruption. Some very generic symptoms like feebleness and diarrhoea, which can be linked to all three causes, are omitted. So is the claim that some patients' tongues fell off (MS), because the generally more reliable report by JS says this was not the case, with one possible exception he had from hear-say only. Sources: letters from OF = Icelandic officials [Wieners20SM], JS = report by Jón Steingrímsson (Steingrímsson, 1788), MS = report by Magnús Stephensen (Stephensen, 1785), RoX = (Roholm, 1937), chapter X.



Sigurdarson and Pálsson (1957) found that $20-40mg/day$ of fluorine for 1/2 year caused mild skeletal fluorosis in sheep. RoXX estimated that a cryolite worker who showed significant skeletal fluorosis (stiffness in the vertebral column) but was otherwise in a reasonable condition, had ingested $15mg/day$ over 25 years. We take $10mg/day$ for 1/2 year as a lower estimate of the critical dose for mild skeletal fluorosis.

During the Laki eruption, people might theoretically have taken up fluorine in various ways: by inhaling of gas (HF) or ash, in contaminated drinking water, through animal-based food or through plant-based food.

*Inhalation of HF.* This is unlikely to be a major source of fluorine, except maybe at close vicinity from the volcano, because the HF concentration in volcanic plumes is typically far below health guidelines (IVHHN, 2022).

*Inhalation of ash.* Dust (ash) of particle diameters up to $100\mu m$ (PM100) can be inhaled. Particles larger than $10\mu m$ are typically intercepted in the upper airways, but even there they may leach fluorine to moist tissue. We estimate daily fluorine ingestion from dust inhalation as follows:

$$F_{inh} = V_{breath} \times C_{PM100} \times M_F \qquad (5)$$

where $V_{breath} \approx 10m^3$ is the volume of air inhaled per day, $C_{PM100}$ the concentration of PM100 in $\mu g/m^3$, and $M_F$ the mass fraction of fluorine in ash. To obtain a generous upper estimate for $C_{PM100}$ we start from the highest daily-mean PM10 concentration measured in Vík after the 2010 Eyjafjallajökull eruption, which was $1230\mu g/m^3$. The grain size distribution of Eyjafjallajökull ash from Thorsteinsson et al. (2012) suggests that the concentration of PM100 was 4 times that of PM10, which yields $C_{PM100} \approx 5000\mu g/m^3$ if larger dust particles are suspended equally well as smaller ones. Locations far from Lakagígar had a far lower ash load (ash layer thickness $< 1mm$ in most of Iceland) and experienced far lower PM100 values than Vík after the Eyjafjallajökull eruption. In addition, probably not all days were windy enough to allow resuspension of old ash. Thordarson and Self (2003) estimate the fluorine content of the Laki ash as $M_F = 550ppm$ (by mass). Inserting these values into eq. 5 yields a daily intake of about $0.03mg$ of fluorine, far below our estimated critical dose for morbidity.

*Drinking water.* Comparisons with Hekla eruptions allow some very rough estimate of fluorine concentration in drinking water. After the 1947 eruption, fluorine concentrations of up to 9.5mg/l were found in nearby rivers (Stefánsson and Sigurjónsson, 1957, cited in Stewart et al. (2006) After the 1970 eruption, rivers were found to contain up to 10mg/l, which diminished to up to 0.5mg/l within 2 weeks, while stagnant water contained 4-70mg/l, reduced to 0.3-14mg/l after 2 weeks (Georgsson and Pétursson (1972), cited in Wilson et al. (2009)). In 2000, a peak concentration of 0.7mmol/kg, or about 14mg/l, was measured in the spring-fed river Ytri-Rangá, after steady rain had flushed the soluble fluorine into the ground water. The next sample taken 5 days later showed greatly reduced fluorine levels. (Higher values in between measurements cannot be excluded.)

For the last eruption, the ash burden near the river's source was $1-10kg/m^2$ (Flaathen and Gíslason, 2007); assuming a fluorine content of 300ppm near the volcano (**?**) suggests a fluorine burden of $300-3000mg/m^2$. For comparison, it has been estimated that the Laki eruption deposited $450mg/m^2$ on average over a $200000km^2$ area (sect. 2.2). However, this deposition was spread over a longer time period, so that peak concentrations were likely lower, especially in regions far from the eruption.

Except for the inhabitants of small islands, Icelanders drank from (small) rivers or springs rather than stagnant pools. Assuming a generous drinking water consumption of $3l/day$, and a fluorine concentration of 14mg/l (highest value measured



near Hekla), the fluorine uptake would be $42mg/day$, far below the lethal dose. Even the maximum concentration observed for stagnant water, $70mg/l$, would result in lower intakes than the lethal dose of $750mg/day$. In addition, it is unlikely that such high concentrations were sustained for months on end, especially further away from the volcano.

*Animal-based food.* Pétursson et al. (1984) states that fluorine does not penetrate into milk, suggesting that dairy products would have been safe to consume. However, RoXIX found moderately mottled teeth in children whose mothers had experienced prolonged exposure to cryolite dust during or before breast feeding. The children's symptoms were consistent with a fluorine content of 1-2mg/l in the mothers' milk. Assuming that cow or ewe milk had similar fluorine concentrations, 4l of milk per day (providing a daily allowance of 2500kcal) or the equivalent in dairy products, would result in a fluorine intake of 8mg/day. RoXXII found no elevated fluorine content in blood, muscles and organs of fluorine-poisoned pigs, except for the kidney (fluorine content elevated by a factor 4). This suggests meat products were safe to consume. Fluorine does accumulate in bones in significant quantities (RoXXII-XXIV). However, it seems that bone broth was not a common food in Iceland, not even as famine food (Jónasson, 1961). In addition, fluorine in bones is embedded the form of fluorapatite, a mineral with poor solubility in water: Wei et al. (2013) found concentrations below $38mg/l$ ($2mmol/l$) in highly acidic solutions, and far lower values otherwise, suggesting that bone broth, if produced, would not have contained lethal fluorine concentrations.

*Plant-based food* is the most uncertain potential source of fluorine. Plants can take up fluorine from the air (Jacobson et al., 1966) and the soil (Hong et al., 2016), with fluorine accumulation differing widely per plant type; in addition, fluorine-rich ash may have been attached to plants. Mild washing can reduce fluorine content in plant leaves (Jacobson et al., 1966). As grazing animals contracted lethal fluorosis from plant-based food, human-edible plants might have contained dangerous quantities of fluorine as well. However, Icelanders' diet was low in plants, although wild plants such as berries, angelica root, and herbs were used. In addition, Iceland moss (cetraria islandica, a lichen) was harvested in significant quantities in some regions; farmers in Eyjafjarðarsýsla tried to live on 'moss and water' in the summer 1783 [Wieners20SM]. Unfortunately, we have not been able to obtain data on the fluorine content of these plants and Iceland moss. It seems plausible that, given the generally low consumption of plants as well as extensive washing typically applied to Iceland moss (Jónasson, 1961, p40), fluorine intake was much lower than that of grazing animals, but we cannot give an upper bound on plant-related fluorine ingestion. Roholm found that animals' conditions quickly improved after reducing the daily fluorine intake. Thus the fact mortality in Eyjafjarðarsýsla only started to rise months after the farmers had switched from moss to meat and (spoilt) grain from the trade post [Wieners20SM] suggests that fluorosis was not the main cause of death there.

To summarise, our rough estimates suggest that it is rather unlikely for humans to have ingested enough fluorine to reach the estimated lethal dose of 750mg/day, although plant-based food is an uncertain factor. However, we cannot exclude that fluorine intake exceeded 10mg/day. Since we deliberately used high estimates of fluorine contents and low estimates of critical doses, the fact that we cannot exclude fluorine-induced morbidity should not be seen as a proof that it did indeed occur on a large scale.





## 9 Conclusions

While traditionally the large death toll in Iceland after the Laki eruption has been attributed to hunger and an increase in
endemic disease (with the latter probably largely caused by the former), two additional hypotheses were formed in the 1970s
to 90s (Grattan et al., 2003, 2005; Blong, 1984; Friðriksson, 1983): First, that air pollution may have directly contributed
to human mortality in Iceland and (northwestern) Europe, and second, that humans in Iceland may have died from fluorine
poisoning. Both hypotheses are sometimes referred to in scientific literature (Schmidt et al., 2011; D'Alessandro, 2006) and
found their way into popular science (Witze, 2016), but have received little scrutiny, especially as regards comparison with
Icelandic contemporary data.

Using county-level demographic data and parish registries (sect. 3), we show that many features of the data, especially
regional discrepancies (sect. 4) can be satisfactorily explained by hunger and diseases, when not just the effect of the eruption
but also local effects such as the weather and availability of alternative food sources are taken into account (sect. 5). However,
the existence of one explanation does not exclude alternative or additional explanations, hence we also discussed the possible
contributions of air pollution and fluorine poisoning.

For air pollution (sect. 7), comparisons with modern eruptions and other pollution events suggest that air pollution has
immediate effects on human morbidity and mortality. In 1783, mortality in northwestern Europe only started to increase more
than a month after the first appearance thick Laki haze, and in Iceland, no increase in mortality is noted during summer-autumn
1783. Furthermore, modelling studies suggest that at least in Europe, concentrations of fine dust and $SO_2$ were too low to have
significant impacts on mortality. Indirect effects, such as pollution increasing the susceptibility to disease, are impossible to
prove or disprove. The weakness of evidence for a major contribution of air pollution to human mortality in 1783 does not
imply that a similar eruption in modern settings would not have serious consequences (Schmidt et al., 2011; Heaviside et al.,
2021).

Regarding fluorine poisoning (sect. 8), described symptoms in humans which were used as evidence for possible fluorosis,
seem to match better with hunger and scurvy, while classic symptoms of fluorosis, such as mottled teeth and spine stiffness,
were not recorded for humans. In addition, even generous upper estimates of fluorine intake via gaseous fluorine compounds,
inhalation of ash, or ingestion via drinking water or meat, are far below even a conservative estimate for the lethal dose.
One remaining uncertainty is the possible ingestion through plant-based food, for which fluorine concentrations could not be
estimated, but plant-based food played only a small role in the Icelandic diet.

While it is well possible that some individuals died of air pollution or fluorine poisoning, there is no compelling evidence
that either made a significant direct contribution to the large death toll in Iceland and elevated mortality in northwestern Europe.
Unless further evidence emerges, both hypotheses should be treated with caution.

*Data availability.* The parish and county level demographic data from Iceland are available under: https://doi.org/10.34894/9YT5BK



*Author contributions.*  C.W. conceived study, G.H. compiled parish and county-level demographic data, C.W. and G.H. analysed demographic
data, C.W. researched and analysed material on direct impact of air pollution and fluorosis and made graphics, both authors wrote manuscript.

*Competing interests.*  The authors have no competing interests.

*Acknowledgements.*  The authors thank M. Schöneich, A. and G. Wieners and C. Wirries for fruitful discussions and comments on an earlier
version of the manuscript.



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
