# Peer review of ""More poison than words can describe": What did people die of after the 1783 Laki eruption in Iceland?"

_EGUsphere, 2023_

## Author Comment (AC1)

***Reply to Reviewer 1***

*I found the paper highly interesting and well researched. I would only put forward to minor comments:*

*1) In line 123, Gunnlaugsson and Rafnsson should be put within brackets, otherwise it sounds like they are the officials in question.*

Thanks, corrected.

*2) Why is the article by Casey et al. from 2019 not cited? (https://www.ncbi.nlm.nih.gov/pmc/articles/PMC6456407/). This is not my field, but it looks like you might want to include it in the discussion.*

The paper by Casey et al. contains rather dubious results.
In particular, the reported mortality rates are totally off the mark, as they claim that "annual male infant mortality rate [in Sweden] ranged from 5 to 19 per 1000 (mean = 7, SD = 2) and the annual female infant mortality rate from 4 to 19 per 1000 (mean = 6, SD = 2)", while in reality the infant mortality in Sweden fluctuated between 163 and 285 deaths pr. 1000 births. We consulted two specialists of 18th century demographic history in Sweden, and they both agreed that the data in the article made no sense. In comparison, the lowest infant mortality rates recorded for Iceland before the 1890s were around 200 per 1000 births. This alone makes us doubt the validity of the study – either there is something wrong with their data, or the data are represented in a confusing manner inviting misunderstanding of what the authors mean.
In addition, if pollution in the summer of 1783 indeed affected gender-dependent birth outcomes, one would expect to see already some effect in 1783, stemming from the children born in the last months of that year, particularly as some studies suggest that such effects mostly occur in case of exposure in the second trimester of pregnancy.
As an additional check on male/female birth ratio, we computed this ratio for our Icelandic data set, with the following outcomes:
- Fraction of male births, 1769-82: mean 50.22% of all births (lowest value: 47.69% in 1772; highest: 52.76% in 1777)
- Fraction of male births, 1783: 50.69%
- Fraction of male births, 1784: 52.63%
- Fraction of male births, 1785: 48.84%

In other words, the fraction of male births in 1783-84 was slightly higher than normal (not lower, as suggested by Casey et al. for Sweden), but within the range of fluctuations of the previous years. Of course the numbers from 1784 onward could be confounded by possible effects of famine (which were absent in Sweden), but the 1783 value also does not suggest a relevant effect for pollution, despite the fact that pollutant concentrations were much higher in Iceland.
Given that the paper's results are clearly unreliable, we decided not to discuss it in our article.

---

## Author Comment (AC2)

*This paper by Wieners and Hálfdanarson is an excellent review and re-assessment of the causes of mortality in Iceland and Europe during and following the 1783-84 Laki fissures eruption. The analytical methods are thorough, robust and replicable. The findings are compelling and thoroughly explored. The paper is well written and structured, overall. I strongly recommend the paper to be published in NHESS after some minor corrections which are listed below.*

Specific Comments
Of the corrections, the most important are:

- *Consistently, the figures are of poor resolution and are hard to read. This especially applies to the maps which are especially challenging. I would strongly suggest that the authors remove all of the 'columned' numbers from the maps. It seems that most of these data are given anyway in the supplementary material. The reader cannot take in this level of information and it would be better for the maps to be used to clearly demonstrate the key points of the figures rather than trying to cram multiple parameters onto one map. The colour shading (blue to brown) is also not intuitive in terms of immediately understanding which means lower/higher values. Please also use a single font (ideally sans serif) and the key should not overlap with the map. See further detail in the list of comments, below.*

We have done the following to resolve these issues and other issues raised by reviewers:
- Moved fig. 1 into the SI and replaced it with a simpler map only depicting the locations of the counties and how we grouped them into regions.
- In fig. 1b, 2c, 5a the numbers were removed from the graph (they are still in the SI) and the full county titles added. Note that the previous fig. 1 is now in the SI.
- The case where the legend overlapped with the southeast coast of the map has been resoved (fig 1b, now SI).
- In fig. 2a, 2b, the data in the figure were however kept, because the information about the timing of deaths (1783 vs 1784 vs 1785) is used a lot in the text and should therefore be present in the main text together with the mapped information. This means that full county titles could not easily be added here. Since fig 2a,b are rather less crowded than the previous fig. 1b and 5a, and are grouped with fig. 2c which now has full county names, we deem this an acceptable compromise between completeness and plot design. In fig. 2a, 2b we shifted some of the number and text boxes to improve readability and, where possible, made the county abbreviations a little longer (e.g. Snæ -> Snæfellsnes).
- In addition, in fig. 1a (now SI) we removed the population numbers, increased the letter size for the county names, and lightly coloured the counties according to their region (Northeast etc), to improve readability.
- The maps themselves already used sans-serif, but we now also made sure that the subplot titles, which we added in latex, are in sans-serif.

Given that the maps had to be plotted semi-manually because there is no suitable base map with historical Icelandic counties, changing the colour scale would mean an outsized amount of work and was therefore not done, seeing that the colour bar clearly indicates which values are high and low.

- *There are parts of the text which start to feel repetitive and are too detailed, particularly in relation to fishing/livestock ownership (see Section 5). Perhaps some of the detail could be moved to Supplementary Material as the 'story' is still compelling without it.*

  See remark below: some repetitions between sect. 5 and previous sections (2.4) were identified and removed.

**Technical corrections**

- *Add subscripts to molecular formulas like $H_2SO_4$ and $SO_2$ throughout the manuscript.*

  Corrected.

- *Line 63 – 'handled poor relief' – what does this mean? Rephrase so that it is more intuitive.*

  Replaced by: were responsible for administering care for the poor.

- *Line 64 – missing a bracket before 'fig. 1a)'?*

  Corrected

- *Figure 1 is hard to read with so much text. My suggestion is to turn these two maps into two separate figures so that they can be larger. I don't understand the key for Figure 1b at all (on the map, not the figure caption). It's not clear what the colours in the pie charts mean and I can't see the left and right columns that are referred to – I think it refers to the numbers but unclear. What are the colours on the map? Presumably they relate to the bar at the bottom right but what is that bar for? What is 'national mean'? All of this is somewhat described in the caption but the map is overly crowded and hard to understand. It's still not clear from the caption which colours in the pie charts refer to which level of fishing access. Even with explanation, the columned numbers are messy and do we really need this much information? Given that the info is already in the Supplementary Material, it does not need to be replicated here. Unless critical to the argument, I would remove Figure 1b or massively simplify it. Finally, it's not clear to me why the county/regional names are different in the two maps. It is incredibly confusing and makes it hard to then follow the names in the paper.*

See remarks above regarding figures.

- *Line 96 – replace the word 'subsidence' with 'gravitational settling and wind circulation'.*

Subsidence is the meteorological term for downward air movement in a high pressure area, however, it is a bit of a jargon and indeed does not include the (slow) gravitational settling. Changed into: „downward air movement and gravitational setting".

> *Line 102 – was (fine) ash only produced in explosive episodes? Coarser ash would also have been produced by lava fountaining and might then have been abraded to finer ash by the wind. But the coarser ash would also have carried fluorine and would have been dispersed over much of Iceland in strong winds and when lava fountaining was at high effusion rates.*

While the suggested processes may have occurred, we have no detailed information about their relative importance, and their relative effect on fluorine distribution. For the purpose of this study, we therefore stick with Thordarson's and Self's estimates. In the paper, we removed the word „Fine" (-> „Ash was produced") to avoid suggesting that only fine ash from explosive episodes contributed.

> *Line 106 – it is not clear if the 450 mg/m2 refers to the mass of ash deposited on average or the mass of fluorine (I think the latter but this only became clear later in the paper). It's also not clear why your figure is 450 mg/m2 when Thordarson and Self calculated 500 mg/(k)m2.*

Indeed, this value refers to fluorine deposition (not ash); now clarified. Thordarson and Self strongly rounded the value; when we recomputed it, we didn't round 450 off to 500.

> • *Table 1 – what are the data in parentheses?*

The absolute values (death and birth counts) are outside parentheses, the percentages within. Clarified in caption.

> • *Figure 2, Figure 5 – again, the numbers are not helpful on these figures. The maps are hard to read and messy. The text is too small to read.*

See remarks above regarding figures.

> • *Line 274/5 – repeat of the word 'were'.*

Corrected

> • *Line 293 – 'Mortality in whole Iceland' doesn't make sense. The whole of Iceland? Same in Fig 4 caption.*

Replaced by "in the whole of Iceland"

> • *Figure 4 would be easier to interpret if the y axis were the same scale for a-i graphs. Fig 4j – I presume that the colours in graph j are the same key as graphs a-k but this isn't explained in the caption.*

We considered using the same y-axis in each graph a-i, but decided against it, because then in the regions with comparatively low mortality peaks, the graphics become too hard

to read. In particular, the peak mortality in "South" was only 1/5 of that in "Northeast", yet one may want to be able to distinguish different causes of mortality.

Yes, the colour scale in plot j is the same. Added to figure caption.

- *Line 308 – 'the mort' should say most*

Corrected

- *Line 376 – from here on there are mentions of 'fugitives'. What is meant by this term? Usually it would mean a person who has escaped from captivity/prison/arrest. Is that the case here? Why were they captive? Do you just mean displaced people?*

We meant displaced people. Corrected.

- *Line 390 – what is 'burthen'? Burden?*

Yes, we meant burden. Corrected.

- *Line 400 – 'hey' should say 'hay'*

Corrected

- *Especially because Figure1 is hard to read, it is really hard to follow the paper with all the mentions of locations. It would be much easier if these county/regional names came with 'signposting' as to their geographic location. This is sometimes done (e.g. mention of southwest regions) but not always.*

As mentioned above, fig. 1 is replaced by a much simpler map with just the county locations and geographic regions, which we hope will provide a clearer reference.

- *Overall section 5 feels repetitive of earlier information; the text becomes too detailed and rather confusing. The final summary paragraph of this section is excellent though – could the whole thing be shortened into a slightly longer summary?*

We respectfully disagree with the suggestion that sect. 5 is repetitive of earlier sections when looking *on the whole*. In particular, sect. 4 establishes where and when people died and presents the presumed causes of death, and sect. 5 interprets these patterns by attempting to explain them in terms of specific causes, hunger and disease, making use of knowledge of the background situation (sect. 2).

However, looking in detail, some fragments can indeed be seen as repetitive, including:

- Repeating the location of the main fishing regions (established earlier in sect. 2)
- Repeating some details about migration (established in sect. 4)
- Repeating some details about the smallpox epidemic (from sect. 4)

In these instances, we removed redundancies except in one or two cases where they were short and we felt that a brief reminder might help the reader.

In addition, we found that some aspects of the story were spread over sections in a not entirely consistent way. In particular, section 4 contained a lengthy piece on the relationship between hunger and endemic disease, which we have shifted into section 5, along with the discussion of the smallpox epidemic. This way, we achieve a clearer sepatation between the descriptive parts (sect. 4) and the interpreting parts (sect. 5).

> *Line 493 and elsewhere - Lava 'streams' is not a commonly used term. Consider replacing with lava flows if this is what you mean.*

Yes, this is what we mean. Corrected.

- *Line 496 – remove 'health' from 'health volcanic pollution'*

Corrected

- *Line 498 – 'mortality Grattan et al' missing punctuation or parentheses of some kind*

Corrected

- *Line 504 – Wakisaka et al. paper – are you sure this paper just looked at SO2 and aerosol? It's impossible to separate these exposures from volcanic ash exposures which are the primary emissions from Sakurajima volcano.*

Thanks for pointing this out. Wakisaka looked at SO2 and total suspended matter, which includes at least sulphate aerosol and (fine) ash. See also response to the comment regarding table 3c.

- *Line 561 – remove the rogue bracket after 'mature'.*

  Corrected

- *Line 569 – 'on 2004' I think this should be 'in 2004'*

  Corrected

- *Table 3a – it is not clear in the table or caption that you are citing country regulations. It would be much better to cite the country authorities (e.g., US EPA) which have the least and most 'strict' guidelines rather than citing IVHHN as the source.*

We changed the title of the table to clarify that we used national guidelines cited in that IVHHN document („recommended max. concentrations in µg/m^3 according to national guidelines cited in [IVHHN]").

- *Table 3, in general – I do not think that these sub-tables qualify as being labelled, together, as Table 3. They are separate and should be separately numbered.*

Unless this violates the formatting rules of the journal, we propose that the tables should remain together to ensure they are not spread over different pages. Ensuring to keep them on the same page helps to improve readability because values from different subtables should be compared (i.e., actual concentrations with thresholds for health impacts).

- *Table 3c – 'IVHHN: Volcanic aerosol mostly in PM2.5 range, thus PM2.5=TSM.' This is not correct – most of the PM from Sakurajima is volcanic ash so is larger than PM2.5. Additionally, see Horwell 2007 for comparison of PM10 and PM2.5 ratios for volcanic ash. https://pubs.rsc.org/en/content/articlelanding/2007/em/b710583p and Hillman et al. for an analysis of Sakurajima ash:* https://doi.org/10.1007/s00445-012-0575-3

Thanks, very good point. We implicitly assumed much of the Sakurajima volcanic aerosol to be sulphate, but this is indeed not stated in the Wakisaka paper (in fact, assuming all suspended matter to be sulphate would imply a very high sulphate/SO2 ratio); much of the aerosol must have been (fine) ash.  We are glad that the reviewer pointed this out.

The review paper by Hillman recommended by the reviewer states that according to previous studies, nearly 100% of ash in an *airborne* sample of Sakurajima ash was PM10. Horwell 2007 in turn suggest that 50% of PM10 from (not airborne) ash samples  is PM4 and about 70% of PM4 is PM2.5 (fig 4a). Assuming these numbers to be representative for particulate matter observed in the data used by Wakisaka, this suggests that their total suspended particle values are roughly PM10 values, and 35% thereof are PM2.5. In other words, if Wakisaka's TSP was all ash, then their values need to be multiplied by 0.35 to obtain PM2.5 valies. If the TSP contained also some sulphate, then PM2.5 was a higher fraction of the PM10, because for sulphate, PM10 ≈ PM2.5. We take PM2.5 = 0.35 PM10 as conservative estimate.

In Europe, volcano-induced aerosol was probably little ash (mild ashfall was observed in some Scottish islands), so the PM values obtained from climate models for sulphate aerosol are probably quite representative for the total volcano-induced PM2.5 increase. Modelled values are around 10-30 µg/m^3 except for Chenet2005 which likely is an overestimate. Wakisaka gives about 20µg(PM2.5)/m^3 as the highest annual mean and about 60µg(PM2.5)/m^3 as the highest daily mean. In other words, Wakisaka finds a similar amount of PM2.5 as is modelled for Europe after the Laki eruption, and Wakisaka finds no significant increase in all-cause morality. Thus the Sakurajima data still suggests that Europe probably also did not experience a significant increase in all-cause mortality, with the caveat that PM2.5 from ash may have different effects from sulphate.

However, Sakurajima is less useful to make conclusions about the lethality (or not) of air pollution in Iceland. First, modelled PM2.5 values from sulphate in Iceland exceed those estimated for Sakurajima; second, sulphate may not even have been the whole story in Iceland as at least some nearby regions may have experienced significant ash exposure which is not included in the climate models we cite.

We will correct this in the article. Other lines of evidence we present against the hypothesis that air pollution caused significant mortality in Iceland remain however intact, therefore we believe our overall conclusion remains intact.

- *Line 507 – replace the word 'chapter' with 'section'.*

Probably referred to line 607. Corrected.

- *Line 686 – should there be '(?)' in the text?*

This was a reference formatting error, corrected.

- *References - Stewart et al. (x2) – please list all co-authors of these papers.*

Corrected

- *There are also several other papers by Grattan's group such as Grattan 1998 and Grattan & Pyatt 1999 which could be relevant to your analysis. Please also see Courtillot 2005.*

We agree that Grattan and his group published many more papers than we cite. However, many of these were book chapters or similar that are not accessible online, and/or summaries or reviews, and/or focus on other aspects of volcanic pollution rather than human mortality.

Of the specific suggestions raised by the reviewer, we were not able to identify „Grattan 1998" either form Grattan's publication list or from looking through referene lists in some of his later papers. If „Grattan and Pyatt 1999" refers to this article: „Volcanic eruptions dry fogs and the European palaeoenvironmental record: localised phenomena or hemispheric impacts?", it is definitely an interesting piece of argumentation, comparing the 1783 haze to other hitherto  mysterious haze events, but it does not discuss human mortality from volcanic pollution, and therefore adds not much extra information to our specific topic. Having re-checked Grattan's output, we still think that with the articles we cited, i.e. Grattan, Durant and Taylor (2003) and Grattan, Rabartin, Self and Thordarson (2005), we cover the articles that present explicitly Grattan's evidence for excess mortality in England and France, respectively, while Grattan and Brayshay (1995) and Durand and Grattan (1999) give a good sample of Grattan's descriptive work on the link between pollution and health symptoms.

As regards Courtillot 2005: The article says rather little on the events of 1783. It first summarises previous work of Grattan and others to establish that the 1783 eruption caused excess mortality in Europe – which we argue in our article to be doubtful – and then goes on to speculate that having dozens of such eruptions but several times larger could lead to mass extinctions. The latter is an intriguing thought but tells us little about the Laki eruption, neither can we say anything about it, for even if it can be proven that the Laki eruption killed nobody outside Iceland, a large number of larger Laki-style eruptions could, obviously, be much more deadly.

---

## Author Comment (AC3)

*The authors pursue in their incredibly compelling paper the question what the cause was of human mortality in Iceland after the Laki eruption of 1783. In a very well-structured manner, they present the socio-economic situation of Iceland and the environmental impact of the eruption, followed by a data and method section. In their analysis part they look at the mortality data and later on how hunger and disease alone may explain the excess mortality in Iceland. After a short excursion about the impacts on the European mortality rate, they discuss the possibility of human death due to air pollution. In the final section they asses the possibility of wide-spread lethal fluorosis in humans, finishing with a concise summary. The paper does address relevant scientific questions within the scope of NHESS as it examines previous research claims from the 1970s on about the contribution of fluorine poisoning towards human mortality after a natural hazard such as the Laki eruption of 1783. Though the tools presented in the paper are well-established, it presents a vast amount of new data on Icelandic mortality for the late 18th century. The historic data is presented up to international standards and the sources have interpreted and criticised according to historiography. The scientific methods and assumptions are outlined clearly and very extensively. The results support absolutely the interpretations, and the authors disprove fairly conclusively the mentioned previous research. All the results can be reproduced thank to the thorough explanation of the methodology, where the used functions are clearly defined, and due to the concretisation of the used data in the supplement.*
*The title of the paper clearly defines the content of the paper, i.e. what did the people die of after the Laki eruption of 1783, except for the that it doesn't mention it mainly concerns the geographical region of Iceland. This could be added in the title.*

Changed title to: „.[…] What did people die of after the 1783 Laki eruption *in Iceland*?"

*The abstract is a complete and concise summary of the research presented in the paper. Both – title and abstract – are targeted to a broad audience, especially working in an interdisciplinary field.*

*There are a few points to be said concerning some figures and their captions. In general, the figures are adequately used to facilitate and enrich the lecture of the paper. However, caption of two figures (Fig. 2 and Fig. 4) goes over the page numbers of pages nine (9) and thirteen (13).*

This formatting issue is expected to be resolved by the journal's typesetting during production stage. Hence no action taken.

*Figures 1), 2a), 2b), 2c) and 5a) have identical issues. Too much information is presented in a single graphic. The numbers of each region could be presented as a separate graphic or a table. For the same images the resolution is subpar and could be enhanced. This would allow the preexisting graphs to show the full county titles, whereas for now the abbreviations are sometimes hard to detect.*

- In fig. 1b, 2c, 5a the numbers were removed from the graph (they are still in the SI) and the full county titles added. Note that the previous fig. 1 is now in the SI.
- In fig. 2a, 2b, the data were however kept, because the information about the timing of deaths (1783 vs 1784 vs 1785) is used a lot in the text and should therefore be present in the main text together with the mapped information. This means that full county titles could not easily be added here. Since fig 2a,b are rather less crowded

than the previous fig. 1b and 5a, and are grouped with fig. 2c which has full county names, we deem this an acceptable compromise between completeness and plot design. In fig. 2a, 2b we shifted some of the number and text boxes to improve readability and, where possible, made the county abbreviations a little longer (e.g. Snæ -> Snæfellsnes).

- In addition, in fig. 1a we removed the population numbers (available in the SI), increased the letter size for the county names, and lightly coloured the counties according to their region (Northeast etc), to improve deadability. Note that the previous fig. 1 is now in the SI.
- Finally, we moved the improved fig. 1 into the SI, seeing that it contains more detail than needed in the main text. Instead, the main text now contains a figure 1 which simply displays the locations and names of the counties and how we grouped them into regions.

*In Figure 3 there is a mistake in the caption: In row 2, where it says "a value of 5 in December 1785 means that from January 1st 1783 till December 31st 1783", it should be called "a value of 5 in December 1785 means that from January 1st 1785 till December 31st 1785"*

Thanks for pointing this out. In fact, the correct version is: "a value of 5 in December 1785 means that from January 1st **1783** till December 31st **1785**, 5 times as many people died as would normally die in one year" as these are cumulative data. Corrected.

*The authors quite clearly give credit to previous work. This is done very precisely in the section where the problem of the previous research is presented. Another good example is the section where the mortality outside of Iceland is discussed by reference to research done by others. The number and quality of references are appropriate. There is no place where complementary references would be required. They are accessible to all scientists.*

*Whereas the overall presentation is very well structured, and the reader is guided towards the conclusions throughline, section 8 about fluorine poisoning might require some more specialized background knowledge to assess the final conclusions. Introduction and summary are supremely well and concisely written. A few other sections contain rather an extensive amount of information. Some part of the methodology section in 3.1. and 3.2 could be shortened and the still crucial but supplementary information could be placed into the annexe. Similarly, towards the end of the paper the methodology section in 8.2 could be pruned slightly and additional information be put likewise in the supplement.*

We carefully re-read sect. 8.2 to check for possibilities to move material into the Supplementary Information. However, we found no satisfactory way to do this – either one could move small bits and pieces, generating a fragmented SI and forcing the interested reader to jump back and forth between the main and the SI, or one would have to move all the quantitative estimates and leave only a brief summary in the main text. This seems to us not a suitable approach, seeing that the quantitative estimates are an essential part of arriving at our conclusions, and that this journal aims at readers who have some interest in numbers. Therefore we propose to leave sect. 8.2 as it is.
We did however shift some index definitions in 3.1 and 3.2 into the SM, leaving only the qualitative descriptions in the main text.

*The technical language is precise and understandable for sicentists from different fields. The English language is of high quality and presentable for a diversified audience.*

*The supplementary material serves deeper understanding and allows the repetition of research especially thanks to vast dictionary of Icelandic terminology.*

**Further notes:**
***Section 1 Line 41-50:***
*Concerning the eruption's climate impact: Just because it is probabilistic that doesn't mean it shouldn't be considered. However, given their focus on mortality statistics not studying climate impact does not detract from the author's conclusions.*

Slightly reformulated to avoid the impression that probabilistic effects should not be considered in general (even though we don't focus on them here).

***Section 4.1 Line 203:***
*Concerning the small Pox epidemic unrelated to Laki: Question: Doesn't hunger or lack of nutrition and migrating population, which the authors relate to Laki, worsen a smallpox epidemic?*

Agree that such an effect cannot be excluded. However:
- Small-pox arrival was external: The disease was not endemic, but could flare up when an  infected ship came to Iceland. This could have happened in any year (a devastating smallpox epidemic occurred in 1707), so the arrival of the disease in autumn 1785 was a coincidence.
- The famine was over by the time the smallpox epidemic started; anecdotal evidence (Steingrímsson 1788) suggests efforts had been taken in summer 1785 to send displaced persons back to their area of origin if they had not found a new residence, so probably the migration and overcrowding had diminished by autumn 1785.

Any conection between the famine and the smallpox would have been less direct than the flare-up of already present endemic diseases which often accompany famines. It is of course impossible to exclude that people were still weakened from their previous ordeal and therefore more prone to illness than they would have been if the same small-pox had arrived five years later or earlier. Conversely, it may be that the famine carried off some weakened or ill persons that would have been particularly susceptible to dying from smallpox, thereby lowering the death rates.
We now briefly discuss this in the paper (end of sect. 5).

***Section 4.2 Line 274-275:***
*Concerning orthography: "were" is repeated*

Corrected

***Section 6.1 Line 400:***
*Concerning orthography: "hey" should be "hay"*

Corrected

---

## Author Response (AR2)

Dear prof. Donovan,

Based on your suggestion we have made the following changes:

*Figure 2 is still hard to read, I'm afraid. I suggest putting the numbers into a small table that forms part of each of 2a and 2b, so that the reader can readily read the numbers and see what they mean. A table for each figure with geog region, then 1783, 1784, 1785 and av would work and be easier to interpret. At the moment the figure is too busy.*

Tables were added as suggested, and numbers were removed from the plots. Since it looked utterly weird to have tables for fig 2a and b, but not 2c, we added a table for that plot too.

*Figure 3 and 4 would benefit from having a single, larger legend (rather than a tiny one on each graph), and larger text.*

Fig. 3 now has a joint legend. Letter type was increased by 25%.
Fig 4a-j now have a joint legend. Letter type was increased by 25%.

*As Reviewer 3 notes, Tables 3a,b,c are all different and should be separately labelled. To keep them on one page, please add a note to the typesetter in a comment and check at proof stage.*

Tables were separated and a comment to the typesetters added to the .tex.

*Please go through the paper and check that column headings in tables are capitalised*

Done!

*Additional changes:*
We corrected a handful of typos and made some very minor textual changes, as can be seen in the track-change document.

Thanks for the helpful suggestions and kind regards
Claudia Wieners and Guðmundur Hálfdanarson